# Spatiotemporal patterns of N₂ fixation in coastal waters derived from rate measurements and remote sensing

Mindaugas Zilius[1]*, Irma Vybernaite-Lubiene[1], Diana Vaiciute[1], Donata Overlingė[1], Evelina Grinienė[1], Anastasija Zaiko[2,3], Stefano Bonaglia[1,4], Iris Liskow[5], Maren Voss[5], Agneta Andersson[6], Sonia Brugel[6], Tobia Politi[1], and Paul A. Bukaveckas[7]*

[1]Marine Research Institute, Klaipeda University, Klaipeda, 92294, Lithuania
[2]Coastal and Freshwater Group, Cawthron Institute, Nelson, 7042, New Zealand
[3]Zealand Institute of Marine Science, University of Auckland, Auckland, Private Bag 92019, New Zealand
[4]Department of Marine Sciences, University of Gothenburg, Box 461, Gothenburg, 40530, Sweden
[5]Department of Biological Oceanography, Leibniz Institute for Baltic Sea Research, Rostock, 18119, Germany
[6]Department of Ecology and Environmental Sciences, Umeå University, Umeå, 90187, Sweden
[7]Center for Environmental Studies, Virginia Commonwealth University, Richmond, VA 23284, USA

*Correspondence to*: Mindaugas Zilius (mindaugas.zilius@jmtc.ku.lt) and Paul A. Bukaveckas (pabukaveckas@vcu.edu)

**Abstract.** Coastal lagoons are important sites for nitrogen (N) removal via sediment burial and denitrification. Blooms of heterocystous cyanobacteria may diminish N retention as dinitrogen (N₂) fixation offsets atmospheric losses via denitrification. We measured N₂ fixation in the Curonian Lagoon, Europe's largest coastal lagoon, to better understand the factors controlling N₂ fixation in the context of seasonal changes in phytoplankton community composition and external N inputs. Temporal patterns in N₂ fixation were primarily determined by the abundance of heterocystous cyanobacteria, mainly *Aphanizomenon flosaquae*, which became abundant after the decline in riverine nitrate inputs associated with snowmelt. Heterocystous cyanobacteria dominated the summer phytoplankton community resulting in strong correlations between chlorophyll-a (Chl-a) and N₂ fixation. We used regression models relating N₂ fixation to Chl-a, along with remote sensing-based estimates of Chl-a to derive lagoon-scale estimates of N₂ fixation. N₂ fixation by pelagic cyanobacteria was found to be a significant component of the lagoon's N budget based on comparisons to previously derived fluxes associated with riverine inputs, sediment-water exchange and losses via denitrification. To our knowledge, this is the first study to derive ecosystem-scale estimates of N₂ fixation by combining remote sensing of Chl-a with empirical models relating N₂ fixation rates to Chl-a.

## 1 Introduction

Biological dinitrogen ($N_2$) fixation plays an important role in the nitrogen (N) budget of aquatic ecosystems as it transforms gaseous $N_2$ into reactive forms, which are available for assimilation by microorganisms, algae, and plants (Gruber, 2004; Hayes et al., 2019). Coastal ecosystems contain diverse diazotrophic communities, comprised of unicellular cyanobacteria, colonial heterocystous cyanobacteria, and heterotrophic bacteria (Riemann et al., 2010; Bentzon-Tilia et al., 2014; Zilius et al., 2020). Unicellular cyanobacteria and heterotrophic diazotrophs dominate in tropical and oligotrophic marine systems (Zehr et al., 2003; Riemann et al., 2010; Farnelid et al., 2016), whereas colonial heterocystous cyanobacteria dominate $N_2$ fixation in temperate systems (Klawonn et al., 2016). The dominant colonial heterocystous cyanobacteria in the Baltic Sea and its coastal areas are *Aphanizomenon*, *Nodularia* and *Dolichospermum* (formerly *Anabaena*) (Olofsson et al., 2020a). Their proliferation has the potential to alter N cycling and thereby influence N export to coastal waters.

Eutrophic lagoons, like those in the Baltic region (Asmala et al., 2017), undergo a shift from phosphorus (P) limitation in spring to N and silica limitation in summer and fall (Patuszak et al., 2005; Vybernaite-Lubiene et al., 2017). Such changes, which largely depend on the timing and magnitude of riverine nutrient inputs, create a temporal niche for diazotrophic cyanobacteria that are capable of overcoming N limitation, and thus often dominate the summer phytoplankton community (Paerl and Otten, 2013; Vybernaite-Lubiene et al., 2017). Large blooms of living cyanobacteria are associated with high oxygen demand in the water column, due to autotrophic and heterotrophic respiration. High oxygen demand results in transient (night-time) hypoxia and enhances the release of dissolved inorganic phosphorus (DIP) from sediments (Zilius et al., 2014; Petkuviene et al., 2016). Thus, a positive feedback is established whereby surplus P favors N-limitation and further proliferation of cyanobacteria that are capable of fixing $N_2$. Although there has been progress in understanding the causes and expansion of cyanobacterial blooms in eutrophic coastal ecosystems (e.g.; Paerl et al., 2001; Paerl and Paul, 2012; Bartoli et al., 2018 and references therein), seasonal patterns and environmental controls of $N_2$ fixation associated with cyanobacterial blooms are not well understood. Particularly challenging is the estimation of $N_2$ fixation at larger scales, owing to the patchy distribution of cyanobacteria (Rolff et al., 2007).

In the present study, we analysed spatiotemporal patterns of pelagic $N_2$ fixation in relation to plankton community characteristics in the Curonian Lagoon. The lagoon is characterized by recurring summer blooms of cyanobacteria, with chlorophyll a (Chl-a) concentrations as high as $400\ \mu g\ L^{-1}$ (Bresciani et al., 2012; Vaičiūtė et al., 2021). Our prior work showed that the occurrence of cyanobacteria blooms had a large effect on N cycling and retention in this system (Zilius et al., 2018). Cyanobacteria blooms reduced annual N retention in the lagoon because summer $N_2$ fixation offsets winter denitrification. In addition, N contained in cyanobacteria biomass was more likely to be exported from the lagoon to the Baltic Sea, rather than buried in sediments, due to their positive buoyancy. A limitation of the prior study was that the patchy and dynamic nature of cyanobacteria blooms resulted in considerable uncertainty in estimating $N_2$ fixation at larger spatiotemporal scales (e.g., for monthly, lagoon-scale N balances). In this follow-up study, we focus on seasonal changes in phytoplankton abundance and

community composition and their utility in predicting $N_2$ fixation. We derive models relating $N_2$ fixation to heterocyst density, abundance of heterocystous cyanobacteria and Chl-a, and use remote sensing of Chl-a to derive estimates of $N_2$ fixation at larger spatiotemporal scales. We consider $N_2$ fixation in the context of other, previously measured inputs and losses of N from the lagoon.

## 2 Material and Methods

### 2.1 Study site

The Curonian Lagoon is located along the southeast coast of the Baltic Sea (Fig. 1). It is the largest coastal lagoon in Europe (area = 1584 km$^2$). The lagoon is a shallow waterbody (mean depth = 3.8 m) that discharges to the Baltic Sea through the narrow Klaipeda Strait and receives inputs from the Baltic during periods of wind-driven forcing (Zemlys et al., 2013). These events are typically of short duration and result in small increases in salinity (typically by 1–2, maximum = 7) in the northern portion of the lagoon. During sampling carried out for this study, salinity in the lagoon was below < 0.5, suggesting limited brackish water intrusions. There was little difference in temperature between surface and the bottom layers (< 2 ºC) indicating well-mixed conditions within water column (Zilius et al., 2020).

The Nemunas River is the principal tributary (mean annual discharge = 16.4 km$^3$) and main source of nutrient inputs to the lagoon (Vybernaite-Lubiene et al., 2018; Zilius et al., 2018). The inflow of the Nemunas enters near the mid-point along the north-south axis of the lagoon. Hydrodynamic modelling studies suggest that the bulk of riverine inputs travel north toward the Klaipeda Strait resulting in shorter water residence time in the northern lagoon (Umgiesser et al., 2016). Longer water residence time in the central and southern lagoon provides favorable conditions for cyanobacteria bloom development (Bartoli et al., 2018). Patterns of phytoplankton seasonal succession are similar throughout the lagoon, transitioning from diatom and chlorophyte dominance in spring to cyanobacteria-dominated blooms in summer and fall (Semenova and Dmitrieva, 2011; Zilius et al., 2018). Spatially-extensive blooms of heterocystous cyanobacteria (*Aphanizomenon* and *Dolichospermum*) are occasionally observed, particularly during low-wind conditions (Bartoli et al., 2018; Vaičiūtė et al., 2021).

### 2.2 Sample collection

We measured pelagic $N_2$ fixation and characterized bacterioplankton and phytoplankton communities at stations located in northern and south-central regions of the lagoon during April–November 2018 (Fig.1). Water samples were collected monthly at two depth layers (0–1.5 m, and 2.0–3.5 m) in the deeper, central site (mean depth = 3.5 m), and at one depth layer (0–1.5 m) in the shallow northern site (mean depth = 1.5 m). Water samples were transferred to 1) sterilized amber borosilicate bottles (0.5 L) for bacteria counting, 2) to opaque HPDE bottles (2 L) for nutrient analyses, and 3) 20 L jars for $N_2$ fixation measurements. All samples were transported on ice (except for $N_2$ fixation experiment) within half an hour after collection for subsequent laboratory processing and analyses. During each sampling, water temperature, salinity and dissolved oxygen were

measured *in situ* at the surface (0.5 m depth) and bottom (0.5 m above the sediment) using a YSI 460 multiple probe (Xylem).

Vertical profiles of photosynthetically active radiation (PAR) were measured with a LI-192 underwater quantum sensor (LI-

COR®). We also monitored total nitrogen (TN) concentrations in the Nemunas River (Fig. 1) to derive riverine N loads for

comparison with atmospheric N inputs via $N_2$ fixation. River samples were collected twice monthly during peak discharge

(January–April) and monthly throughout the rest of the year (16 collections). Water samples (2 L) were collected in triplicate,

integrating the whole water column with repeated Ruttner bottle sampling at the surface (0.4 m depth) and bottom layers (3.0

m depth) as described in Vybernaite-Lubiene et al. (2018). Integrated water samples were transferred to opaque bottles (2 L),

cooled with ice packs, and transported to the laboratory within the hour for subsequent analyses (see section 2.3 for details).

Riverine N concentrations were used in combination with daily discharge measurements (provided by Lithuanian

Hydrometeorological Service) to derive monthly N loads to the lagoon as previously described in Zilius et al. (2018). μ

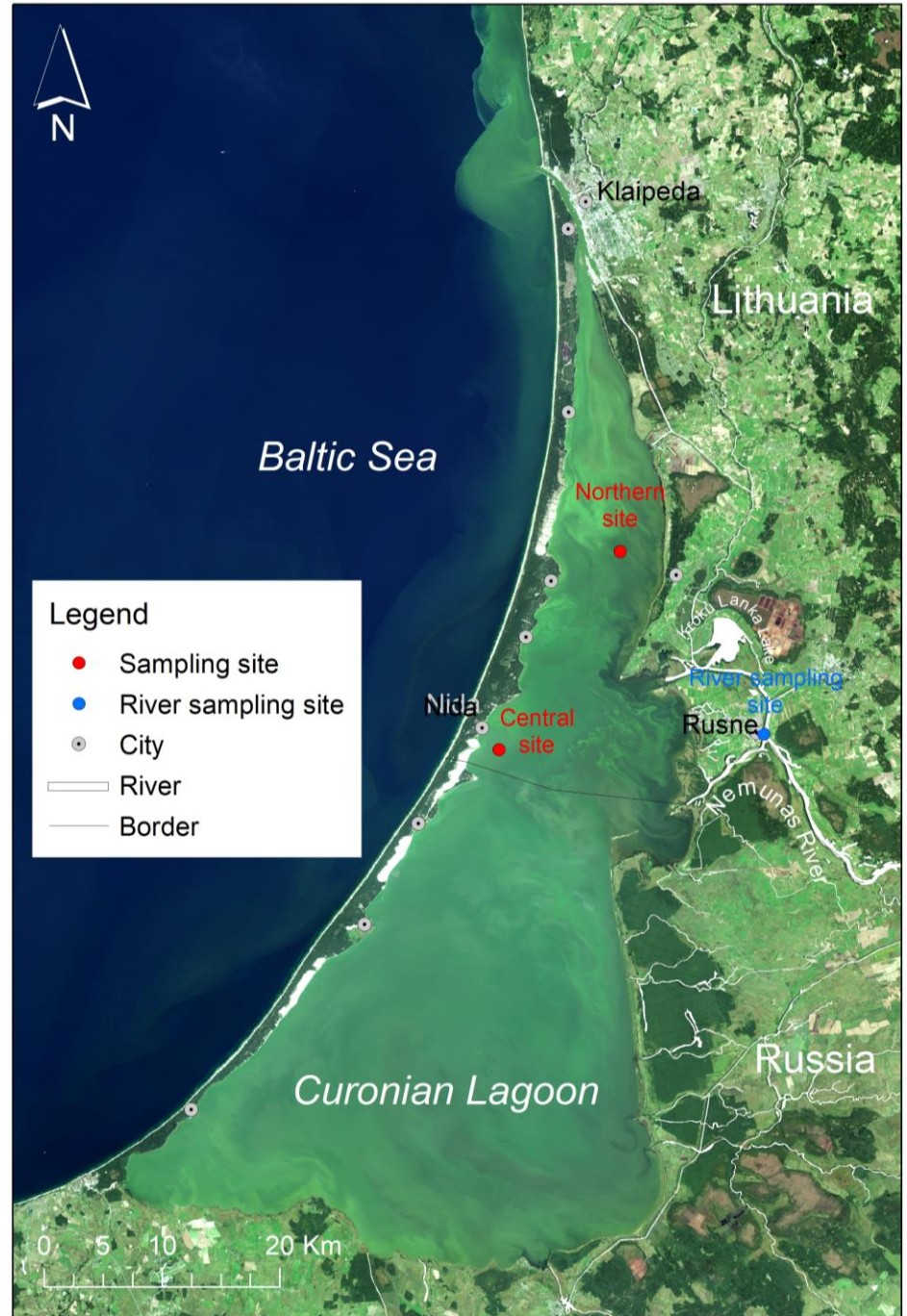

**Figure 1:** Satellite image by OLI/Landsat-8 (18/09/2014) showing summer blooms in the Curonian Lagoon with the sampling sites (red circles) representing the northern and south-central regions, and monitoring site at the Nemunas River (blue circle). LT = Lithuania, RUS = Russia, and black line indicates a border between two countries.

## 2.3 Chemical analysis

Triplicate water sample from each site and layer were filtered (Whatman GF/F, pore size 0.7 µm) for inorganic and organic nutrient analysis as previously described by Vybernaite-Lubiene et al. (2017). Dissolved inorganic nutrients ($NH_4^+$, $NO_2^-$, $NO_3^-$, and DIP) were determined colorimetrically using a continuous flow analyser (San$^{++}$, Skalar) following the methods described in Grasshoff et al. (1983). Total dissolved nitrogen (TDN) was analyzed by the high temperature (680 ºC) combustion, catalytic oxidation/NDIR method using a Shimadzu TOC 5000 analyzer with a TN module. Dissolved organic nitrogen (DON) was calculated as difference between TDN and DIN ($NH_4^+ + NO_2^- + NO_3^-$). Dissolved organic carbon (DOC) was determined with a Shimadzu TOC 5000 analyzer using an acetanilide dilution series as a standard (Cauwet, 1999). Total dissolved phosphorus (TDP) was determined after digestion and oxidation of the organic P forms with alkaline peroxodisulphate acid digestion (Koroleff, 1983). Dissolved organic phosphorus (DOP) was calculated as difference between TDP and DIP. Water samples for Chl-a were filtered through GF/F filters. Pigments were extracted with 90% acetone (24 h at 4 °C) and measured by spectrophotometry (Jeffrey and Humphrey, 1975; Parsons et al., 1984). Particulate matter was collected on pre-ashed (4 h at 550 ºC) Advantec GF75 filters (nominal pore size 0.3 µm) for particulate nitrogen (PN, river water) and isotopic signature ($\delta^{15}$N-PN, lagoon) analysis. Prior to analysis, filters were dried at 60 ºC to constant weight and later analysed with an Elementar Vario EL Cube (Elementar Analysen systeme GmbH) interfaced to a PDZ Europa 20-20 isotope ratio mass spectrometer (IRMS, Sercon Ltd) at the Stable Isotope Facility of UC Davis, USA. The long-term standard deviation is <0.3 ‰ for $\delta^{15}$N. TN was calculated as a sum of dissolved and particulate fractions.

## 2.4 Phytoplankton and bacteria counts

Samples for phytoplankton counting were immediately preserved with acetic Lugol's solution and examined at magnifications of ×200 and ×400, using a LEICA DMI 3000 inverted microscope. Phytoplankton community composition was determined using the Utermöhl method (Utermöhl, 1958) according to HELCOM recommendations (HELCOM, 2017). Phytoplankton biomass (mg L$^{-1}$) was calculated according to the methodology described in Olenina et al. (2006) and converted into carbon units (µg C L$^{-1}$) following Menden-Deuer and Lessard (2000). The number of heterocysts (cells L$^{-1}$) and their frequency per millimeter of filament (mm$^{-1}$) was also determined.

The abundance of heterotrophic bacteria and picocyanobacteria was determined in filtered samples (50 µm size mesh), which were preserved in 0.25% glutaraldehyde (final concentration) and stored at −80 °C until analysis (Marie et al., 2005). Samples for determination of heterotrophic bacteria biomass were stained with SYBR Green I (Invitrogen) to a final concentration of 1:10000 (Marie et al., 2005), diluted with Milli-Q water, and analysed with a flow cytometer (BD Accuri$^{TM}$C6, DB Biosciences) at a medium flow rate (35 µL min$^{-1}$) for 2 min. Microspheres of 1 µm (Fluoresbrite plain YG, Polysciences) were added to samples as an internal standard. Samples for picocyanobacteria (≤ 3 µm) were analysed at a flow rate of 66 µL min$^{-1}$ with an acquisition time of 2 min. Microspheres of 3 µm (Fluoresbrite plain YG, Polysciences) were used as an internal

standard. A factor of 20 fg C per cell was used to convert bacteria counts to carbon biomass ($\mu$g C L$^{-1}$; Lee and Fuhrman, 1987).

## 2.5 Nitrogen fixation

Monthly measurements of N$_2$ fixation in the water column were performed using the $^{15}$N$_2$ technique described in Montoya et al. (1996). Rates were measured in two depth layers (0–1.5 m, and 2.0–3.5 m) at the deeper, central site, and for the whole water column (0–1.5 m) at the shallow northern site. The samples were filled into 500 mL transparent HDPE bottles and carefully sealed preventing formation of air bubbles. Each sample received 0.5 mL $^{15}$N$_2$ (98% $^{15}$N$_2$, Sigma-Aldrich) injected by syringe through a gas-tight septum, and then gently mixed for 10 min (Zilius et al., 2018). As the isotopic equilibration takes up to several hours (Mohr et al., 2010), we incubated the samples for 24 h, thus minimizing equilibration effects which can lead to underestimation of rates (Mulholland et al., 2012; Wannicke et al., 2018). Surface water samples were incubated outdoors at ambient irradiance, while samples from 2.0–3.5 m were wrapped in aluminium foil as *in situ* irradiance was below 1% of surface PAR at these depths (< 5 $\mu$mol m$^{-2}$ s$^{-1}$ in the period of June–November). However, such fixed dark conditions is less representative to *in situ* conditions as cyanobacteria colonies can migrate upward to surface photic zone or use limited light for photosynthesis. After incubation, suspended material was collected on pre-ashed (4 h at 450 ºC) Advantec GF75 filters (nominal pore size 0.3 $\mu$m). This nominal pore size filter was used instead the conventional (0.7 $\mu$m) as it allows for quantitative collection of smaller cells comprising the active diazotrophic community (Bombar et al., 2018). All samples were stored frozen until analysis with a continuous-flow IRMS (Delta S, Thermo-Finnigan) at the Leibniz Institute for Baltic Sea Research Warnemünde (IOW). Volumetric rates of N$_2$ fixation were calculated following Eq. 1 (Montoya et al., 1996):

$$\text{N}_2 \text{ fixation rate } (\mu\text{mol N L}^{-1}\text{d}^{-1}) = \frac{V}{2}\overline{PN} \approx \frac{V}{2} \times \left( \frac{[PN]_0 - [PN]_f}{2} \right) \tag{1}$$

Where V (the specific rate of N$_2$ uptake) is derived from Eq. 2:

$$V = \frac{1}{\Delta t} \times \frac{\left( A_{PN_f} - A_{PN_0} \right)}{\left( A_{N_2} - A_{PN} \right)} \tag{2}$$

$A_{PN}$ is the $^{15}$N atom % enrichment of the PN pool at the beginning ($t_0$) and end ($t_f$) of an incubation, $A_{N2}$ is the $^{15}$N atom % enrichment of the dissolved N$_2$ gas in the incubated water, and PN is the concentration of PN at the beginning ($t_0$) and end ($t_f$) of the incubation. As we have used theoretical estimation of $^{15}$N$_2$ gas dissolution in bottles instead quantification with membrane inlet mass spectrometer, it can result in some underestimation of rates (White et al., 2020). Volumetric N$_2$ fixation rates were converted to areal rates (mmol N m$^{-2}$ d$^{-1}$) taking into account the depth of the water column and the thickness of each depth layer (see above).

## 2.6 Remote sensing

Prior studies have developed and validated techniques for remote sensing of Chl-a in the Curonian Lagoon (Bresciani et al., 2014; INFORM, 2016; Riddick et al., 2019). The satellites can observe the water down to one optical depth, the portion of the

water column where approximately 90% of the remote sensing observed signal originates (Gordon and McCluney, 1975; Werdell and Bailey, 2005). The optical depth of the Curonian Lagoon is typically less than 0.5 m (Vaiciute, unpublished). We obtained satellite images for 6 dates spanning the period when phytoplankton communities were dominated by heterocystous cyanobacteria and water temperatures exceeded 15 $^{\circ}$C (July–September). Chlorophyll-a concentrations were obtained from optical satellite images using an on-board Multispectral Instrument (Sentinel-2: September 7 and 20) and an on-board Ocean and Land Colour Instrument (Sentinel-3: July 4 and 24, August 8 and 23). Satellite images were resampled to a nominal pixel size of 300 m resulting in a grid matrix of ~17,000 cells comprising the area of the lagoon. Cells adjacent to the shoreline were excluded from analyses due to potential interference from aquatic vegetation and benthic algae. Atmospheric correction was carried out using the Second Simulation of the Satellite Signal in the Solar Spectrum-Vector code (6SV; Vermote et al., 1997) previously used in other satellite applications for the Curonian Lagoon (Bresciani et al., 2014). The parametrization of the 6SV code was performed using the Maritime model. Values of Aerosol Optical Thickness (AOT) were obtained from AERONET sites and using MODIS-derived AOT values available from the NASA Giovanni application (https://giovanni.gsfc.nasa.gov/giovanni/). Chl-a concentrations were derived after the application of a semi-empirical band-ratio model that uses reflectance in the red and near-infrared spectral regions (Gitelson et al., 2007; De Santi et al., 2019).

Equation 3 used for Sentinel-2 data:

$$Chl-a, mg\ m^{-3} = (76.36 \pm 2.29) \times \frac{Ref_{705}}{Ref_{665}} - (51.57 \pm 0.26), \tag{3}$$

Equation 4 used for Sentinel-3 data:

$$Chl-a, mg\ m^{-3} = (52.19 \pm 1.81) \times \frac{Ref_{708}}{Ref_{665}} - (32.07 \pm 0.57), \tag{4}$$

where $Ref_x$ indicates the reflectance of the band with central wavelength $x$.

Prior work validating satellite-derived Chl-a against *in situ* observations in the Curonian Lagoon showed good agreement for both Sentinel-2 ($R^2 = 0.91$, root-mean-square error (RMSE) = 18.6 mg m$^{-3}$) and Sentinel-3 $R^2 = 0.95$, RMSE = 7.4 mg m$^{-3}$) images (INFORM, 2016; Riddick et al., 2019). Estimates of $N_2$ fixation were derived for each of the grid cells based on satellite-derived Chl-a and regression models relating $N_2$ fixation measurements to concurrent *in situ* measurements of Chl-a (regressions provided in Results).

**2.7 Statistical analysis**

Principal coordinates analysis (PCoA) was performed to visualize spatiotemporal patterns in plankton community variables (*Aphanizomenon, Dolichospermum,* non $N_2$-fixing cyanobacteria, and heterotrophic bacteria biomass) and their relationship to nutrient concentrations. This analysis was performed using Primer 7 software (v.7, Primer-E Ltd.; Clarke and Gorley, 2015) on Euclidean distances of normalized and forth-root transformed variables. The linear regression was used to predict $N_2$ fixation rates based on *in situ* Chl-a concentration. In addition, variance analysis (two-way ANOVA) was used to test

differences in Chl-a concentration between surface and bottom layers. The assumptions, data normality, and homogeneity of variance, were checked using Shapiro–Wilk test and Cochran's test, respectively. For significant factors, post hoc pairwise comparisons were performed using the Student-Newman-Keuls (SNK) test. The significance level was set at $\alpha = 0.05$. Analyses were performed in SigmaPlot 14.0 software.

## 3 Results

### 3.1 Phytoplankton and bacteria communities

Seasonal patterns in phytoplankton biomass and community composition followed expected trends based on prior work at this site (Fig. 2). *In situ* surface Chl-a during April–July ranged from 25 to 57 µg L$^{-1}$ at the central site and from 14 to 26 µg L$^{-1}$ at the northern site. Higher Chl-a was observed in late summer with values ranging from 52 to 286 µg L$^{-1}$ at the northern site and from 96 to 256 µg L$^{-1}$ at the central site. Phytoplankton biomass showed corresponding changes, increasing from ~1000 to 4000 µg C L$^{-1}$ during spring to late summer. Diatoms dominated the spring phytoplankton community (April–May) accounting for up to 94% of total biomass. During June–July, diatoms were replaced by non and N$_2$-fixing cyanobacteria; the later accounted for up to 36% of total phytoplankton biomass. The non N$_2$-fixing cyanobacteria were dominated by *Planktotrix agardhii* and *Microcystis* spp. Between August and November, N$_2$-fixing cyanobacteria dominated the community (86% of total biomass). The main N$_2$-fixing cyanobacteria were *Dolichospermum* spp. and *Aphanizomenon flosaquae*. Heterotrophic bacteria accounted for ~30% of the total plankton biomass (bacteria + phytoplankton) during the early successional (diatom-dominated) phase. Non filamentous colonial cyanobacteria, such as *Aphanocapsa* spp., *Aphanothece* spp., *Merismopedia* spp. and *Cyanodictyon* spp. exhibited low biomass (< 2% of total) except in June, when their contribution reached 12% at the northern site (Fig. 2c). Picocyanobacteria were not detected during the study period at either site. Heterotrophic bacteria biomass was higher during the cyanobacterial bloom (June–October), increasing from ~80 to 250 µg C L$^{-1}$, however, their relative contribution to total plankton biomass decreased to ~17%. Phytoplankton biomass and community composition were generally similar between surface and bottom layers (April–September), except in October–November when the abundance of N$_2$-fixing cyanobacteria was greater in the surface layer (2500–3500 µg C L$^{-1}$) relative to the bottom layer (< 100 µg C L$^{-1}$).

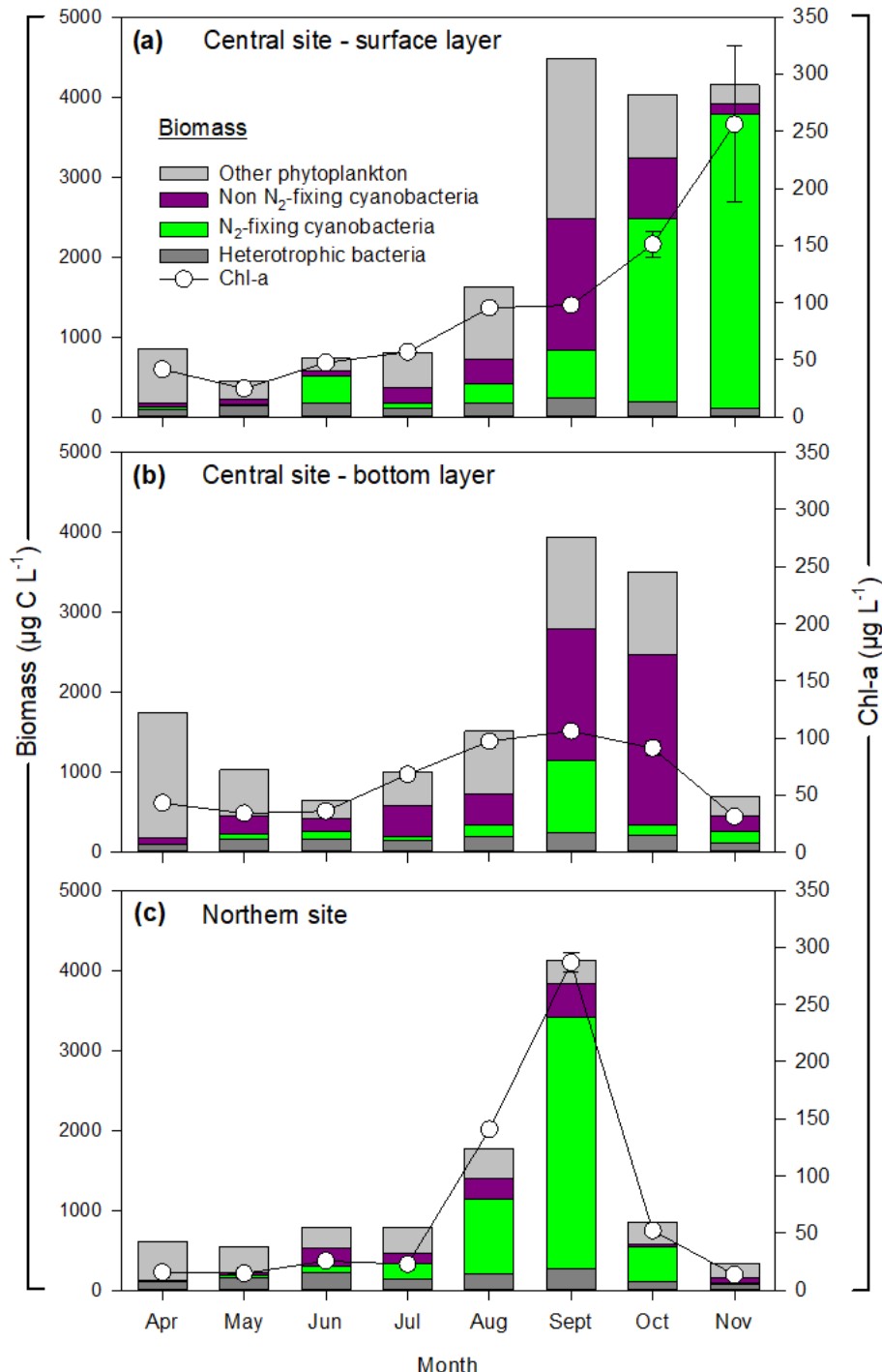

**Figure 2: Phytoplankton and heterotrophic bacteria biomass at central (a, b) and northern (c) sites in the Curonian Lagoon during 2018. Chlorophyll-a concentrations are mean values and standard error (some error bars not visible) based on three replicates.**

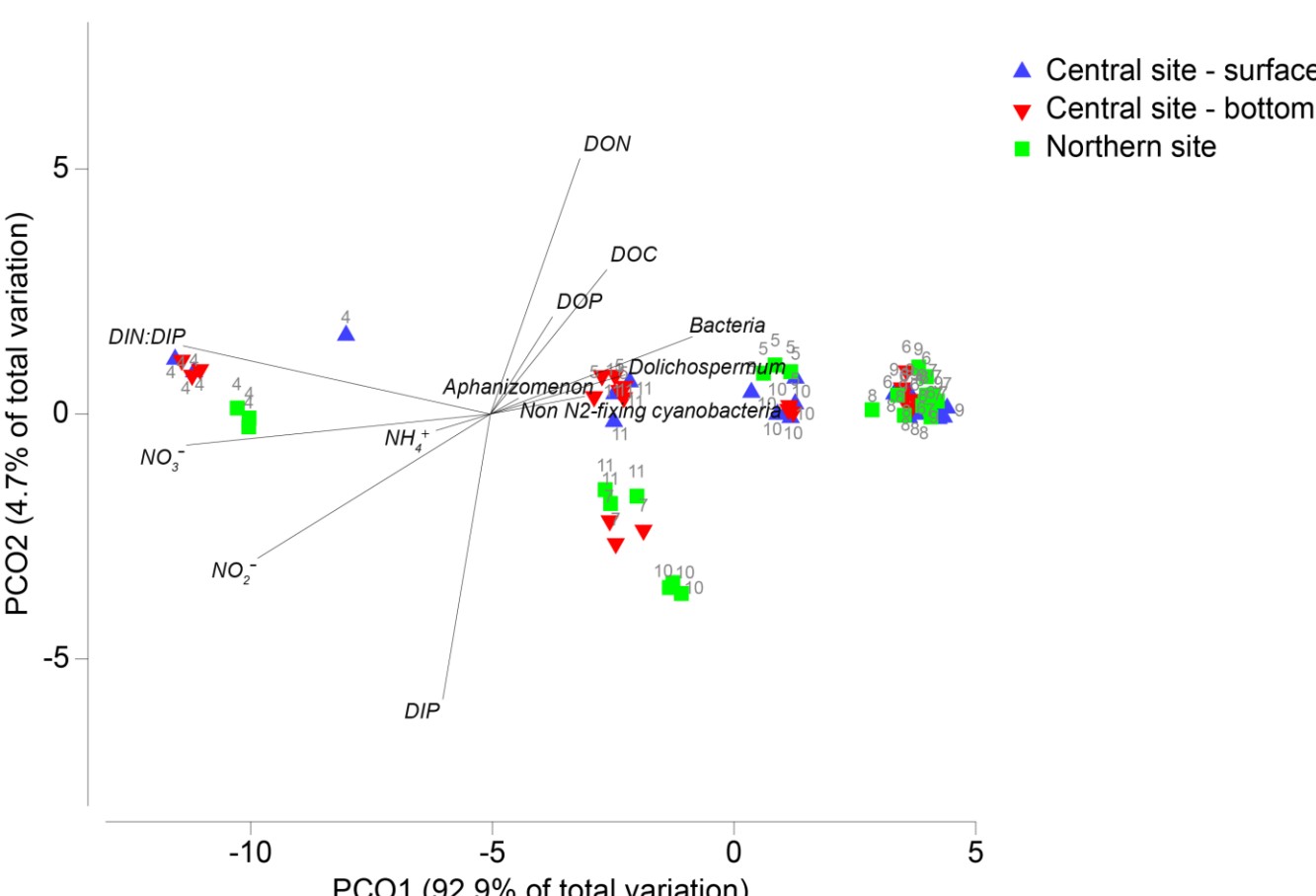

**Figure 3: Principal coordinate biplots generated on Euclidean distances of normalized and forth-root transformed nutrient concentrations (DOC, NH$_4$$^+$, NO$_2$$^-$, NO$_3$$^-$, DON, DIP, DOP, and DIN:DIP). Overlaid vectors show individual chemical variables (those significantly correlating with either of the two primary axes, with Pearson correlations > 0.5) and plankton community biomass (*Aphanizomenon, Dolichospermum,* non N$_2$-fixing cyanobacteria and heterotrophic bacteria).**

Principal coordinates analysis revealed spatiotemporal differences in nutrient concentrations and plankton community characteristics (Fig. 3). The first principal coordinate axis explained 93% of the total variation by differentiating April samples on the basis of high NO$_3$$^-$ (~80 µM) and high DIN:DIP (~200 molar) relative to samples collected in May–November (NO$_3$$^-$ mean = 2.5 µM; DIN:DIP mean = 14 molar; Fig. 3, 4). Biomass of *A. flosaquae, Dolichospermum* spp*.,* non N$_2$-fixing cyanobacteria and heterotrophic bacteria were positively associated with axis 1 indicating their dominance during low N and low DIN:DIP conditions. The second principal coordinate axis accounted for 5% of variation and separated samples collected in July from the bottom layer (central site) and during October–November at the northern site. These samples were

characterized by higher DIP (up to 2 µM) and lower DON (< 40 µM) relative to samples collected at other sites and dates (DIP < 0.5 µM; DON 40–60 µM), but did not reveal differences in community characteristics. Overall, $NO_3^-$ was the dominant fraction of dissolved N in spring, whereas DON was the main fraction in summer and fall. Seasonal variation in DIP and DOP was small (< 2 µM) in comparison to N, such that changes in the relative availability of N vs. P were largely determined by N concentrations.

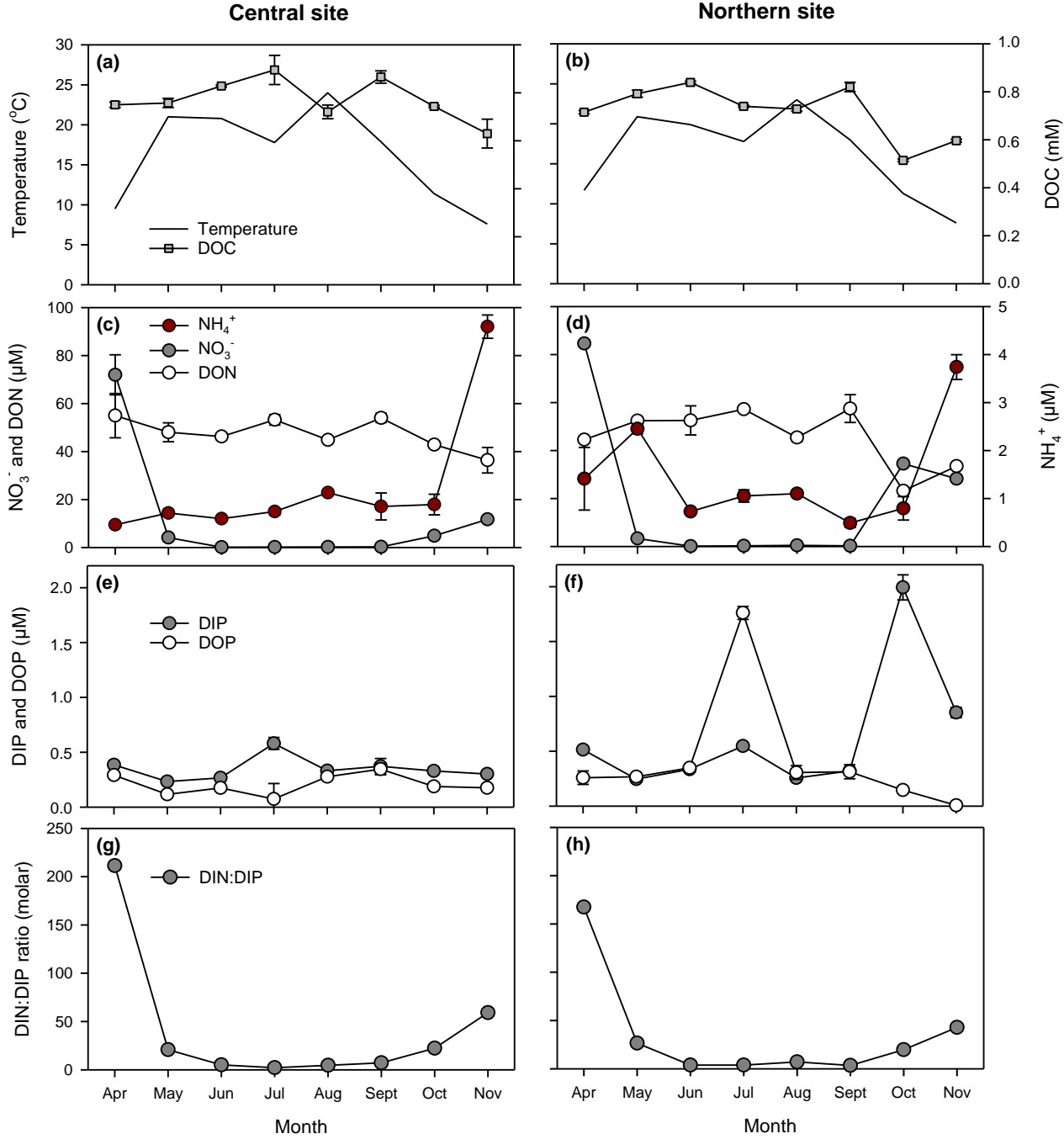

**Figure 4: Temporal patterns in temperature, dissolved organic carbon (a, b), dissolved inorganic and organic nitrogen (c, e), phosphorus (e, f), and DIN:DIP ratios (g, h) at central (surface layer; left panel) and northern (right panel) sites in the Curonian Lagoon during 2018 (error bars denote standard error based on 3 replicates; some not visible).**

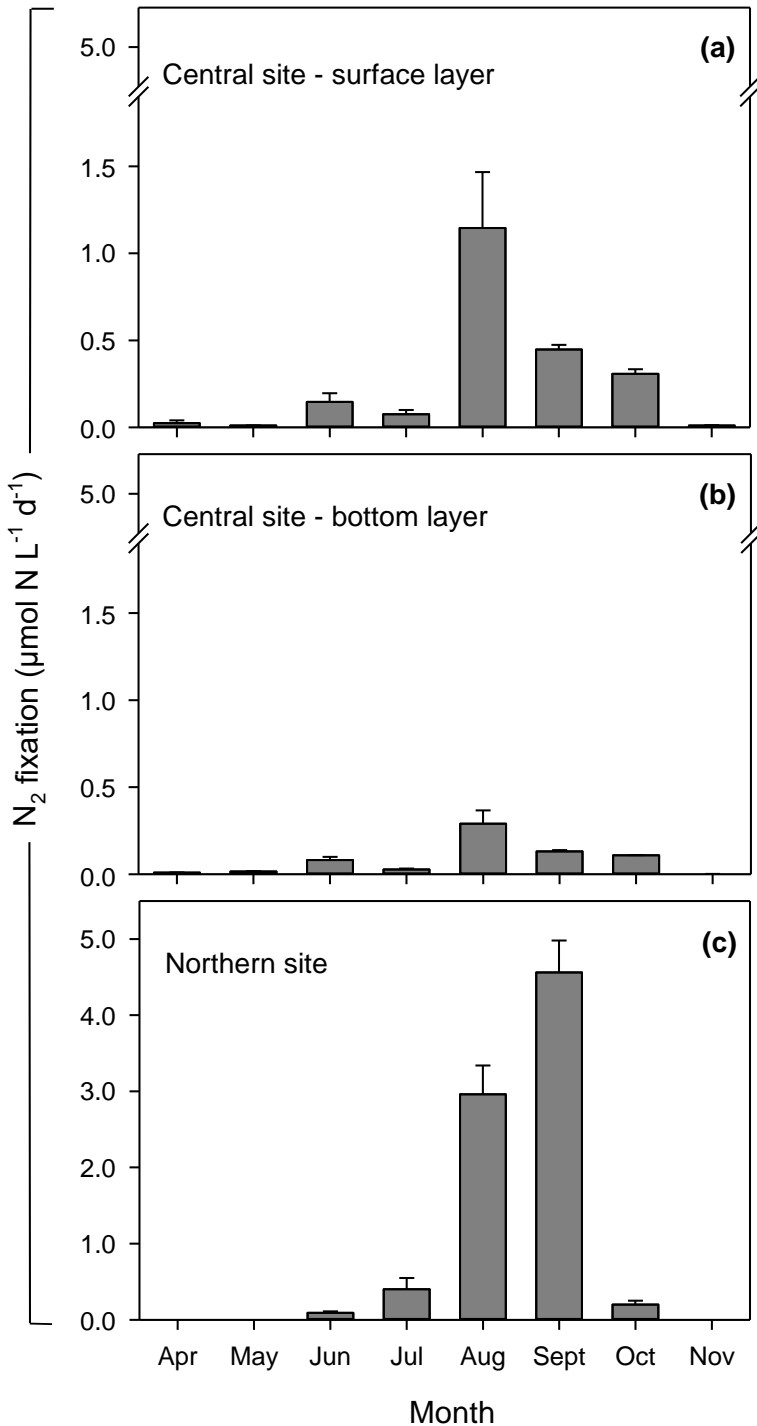

**Figure 5: Rates of N$_2$ fixation at central (a, b) and northern (c) sites in the Curonian Lagoon in 2018 (data are mean values and standard error based on 3 replicates).**

## 3.2 N₂ fixation

Rates of N$_2$ fixation varied by over two orders of magnitude ($< 0.1$ to $> 5$ µmol N L$^{-1}$ d$^{-1}$) with the highest rates measured at
the northern site in August and September ($3.0 \pm 0.4$ and $4.6 \pm 0.4$ µmol N L$^{-1}$ d$^{-1}$, respectively; Fig. 5). A comparison of N$_2$
fixation in the surface and bottom layers at the central site showed that rates were consistently lower ($<0.4$ µmol N L$^{-1}$ d$^{-1}$) in
the deeper layer. The abundance of heterocysts varied seasonally depending on the number of vegetative cells of N$_2$-fixing
cyanobacteria ($y = 0.0251x + 75.0$, $R^2 = 0.92$) with lowest values less than 2000 cells L$^{-1}$ and peak values exceeding 2 million
cells L$^{-1}$ in late summer (Fig. 6a, b). Heterocysts of *A. flosaquae* accounted for $> 95\%$ of total heterocysts as contributions
from *Dolichospermum* spp. were small by comparison ($< 150,000$ cells L$^{-1}$). The heterocyst frequency per filament showed
distinctive temporal patterns between sites depending on the species (Fig. 6c, d). At the central site, two peaks up to 8.0 mm$^{-1}$ in heterocyst frequency of both species was observed during June–September. Whereas heterocyst frequency at the northern
site remained quite high through summer primarily contributed by *A. flosaquae* (~10 mm$^{-1}$), later followed by *Dolichospermum*
spp. (~8 mm$^{-1}$). The N isotopic signature of PN declined with increases in heterocyst frequency and rates of N$_2$ fixation. Prior
to the cyanobacteria bloom, δ$^{15}$N values were 6–9 ‰ and declined to less than 1 ‰ by August before rebounding in October–
November (Fig. 6c, d). Smaller declines in δ$^{15}$N values were observed at the southern site.

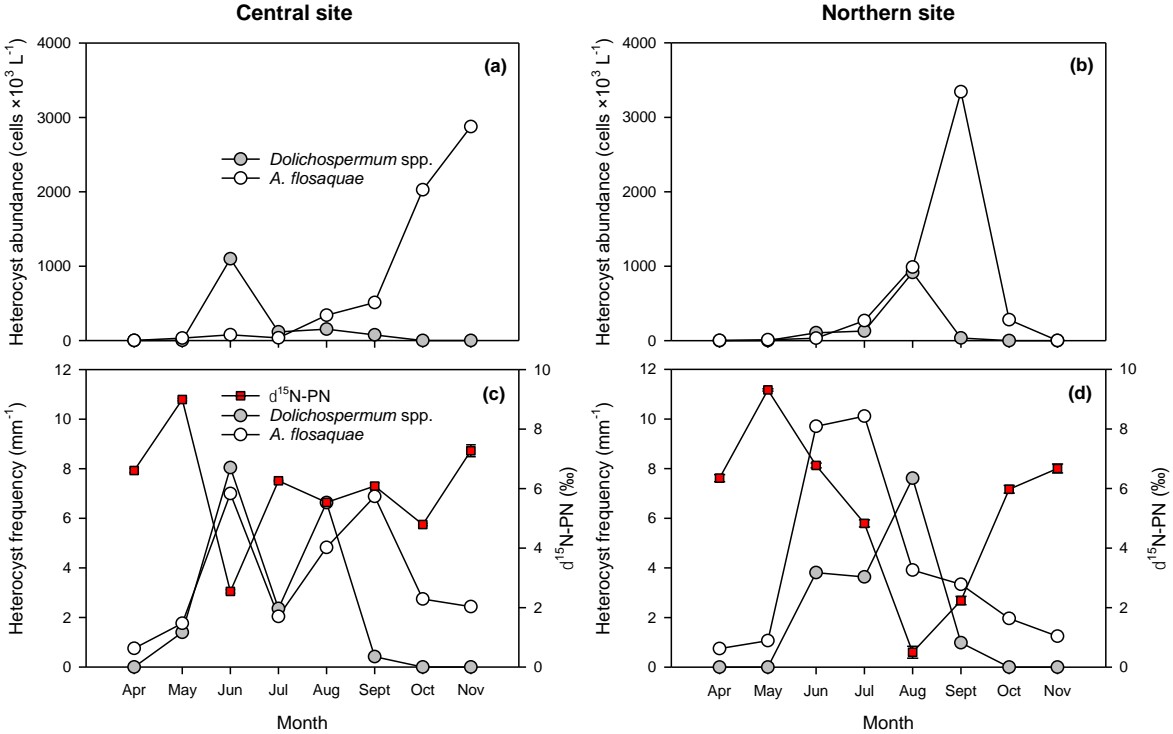

**Figure 6: Abundance of heterocysts of *Dolichospermum* spp. and *A. flosaquae* (a, b), heterocyst frequency per filament length and stable isotope composition of particulate nitrogen (δ$^{15}$N-PN) (c, d) at central (surface, left panel) and northern sites (right panel) in the Curonian Lagoon during 2018. δ$^{15}$N-PN values are mean and standard error based on 3 replicates (some error bars not visible).**

N$_2$ fixation in the surface layer was significantly (p < 0.001) predicted by *in situ* Chl-a concentration (R$^2$ = 0.91), *A. flosaquae* biomass (R$^2$ = 0.83) and *A. flosaquae* heterocysts (R$^2$ = 0.88, all p < 0.001; Fig. 7). Whereas N$_2$ fixation in the bottom layer was weakly explained by *in situ* Chl-a (R$^2$ = 0.52, p = 0.08), but not *A. flosaquae* biomass or heterocysts. *In situ* chlorophyll-specific N$_2$ fixation derived from regression equations (Fig. 7a, b) was considerably lower in the deeper layer (0.002 ± 0.001 µmol N µg$^{-1}$ Chl-a d$^{-1}$) relative to the surface layer (0.018 ± 0.002 µmol N µg$^{-1}$ Chl-a d$^{-1}$).

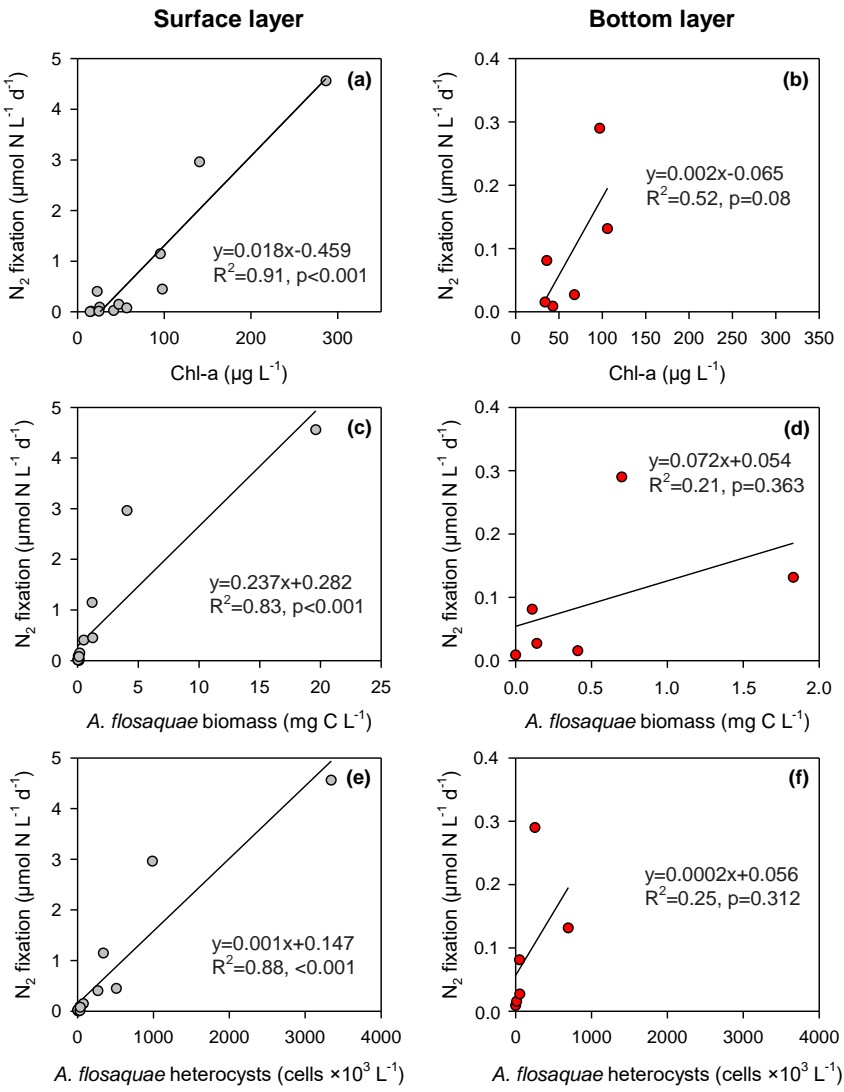

**Figure 7: Relationships between N$_2$ fixation and *in situ* measured Chlorophyll-a (a, b), *A. flosaquae* biomass (c, d) and their heterocysts (e, f) in surface (northern and central sites, left panel) and bottom (central site, right panel) layers of the Curonian Lagoon during April–September 2018.**

Remote sensing-based estimates of lagoon-wide Chl-a increased from $65.4 \pm 0.9 \, \mu g \, L^{-1}$ (July 4) to $88.7 \pm 0.2 \, \mu g \, L^{-1}$ (August 8), and thereafter remained relatively stable throughout August and September (means = 84 to 89 $\mu g \, L^{-1}$). Satellite-derived Chl-a values for each grid cell were used along with the regressions relating $N_2$ fixation to Chl-a to derive estimates of $N_2$ fixation for each cell. We used the relationship between $N_2$ fixation and *in situ* Chl-a (y = 0.018x – 0.459; Fig. 7a) for the surface layer to derive estimates of $N_2$ fixation for the upper water column (0–2 m). For deeper areas, we used the relationship between $N_2$ fixation and *in situ* Chl-a for the bottom layer. We assumed that surface Chl-a (from remote sensing) was representative of Chl-a in the deeper layer as we did not find significant differences between the two layers sampled at the central site (SNK test, $p < 0.05$). The impact of the bloom on $N_2$ fixation can be visualized from the relatively low and uniform rates throughout the lagoon during July, and the subsequent development of localized hotspots in the southern lagoon during August and September (Fig. 8). Lagoon-wide average values of $N_2$ fixation increased from 1.5 (July 4), to 2.5 (August 8) and thereafter remained ~2.3 mmol $m^{-2} \, d^{-1}$ through the end of September.

## 4 Discussion and Conclusion

We characterized seasonal variation in phytoplankton communities in relation to nutrient conditions to better understand the mechanisms regulating pelagic $N_2$ fixation in the Curonian Lagoon. Findings based on this study and our prior work (Zilius et al., 2018) suggest that the decline in riverine $NO_3^-$ inputs following spring snowmelt, and the subsequent depletion of DIN in the lagoon, provides favourable conditions for an active diazotrophic community during summer and fall. Stoichiometric ratios of dissolved inorganic nutrients are frequently used to identify potential limiting elements and their role in driving community succession (Ptacnik et al., 2010; Perez et al., 2011). The occurrence of elevated $NO_3^-$ concentrations and high DIN:DIP after spring runoff was followed by an extended period (8 months) of persistent low N availability, creating a temporal niche for heterocystous cyanobacteria (Supplement, Fig. S1). In a related study, we used molecular techniques to document the diversity of diazotrophs of the Curonian Lagoon and found that the community shifted from $N_2$-fixing heterotrophic bacteria in spring to photosynthetic heterocystous cyanobacteria in summer–fall (Zilius et al., 2020). Though sequences were also attributed to diazotrophic picocyanobacteria (*Synechococcus*, *Crocosphaera*, *Rippkaea*, and *Cyanothece*), these were not detected with flow cytometry, suggesting low abundance. Here, we show that spatial and temporal variation in rates of $N_2$ fixation were primarily determined by the abundance of heterocystous cyanobacteria. The maximum abundance of heterocysts occurred during the bloom of *A. flosaquae* and coincided with the peak in $N_2$ fixation rates and the decline in $\delta^{15}$N-PN to values similar to atmospheric N. The heterocyst frequency per filament of *A. flosaquae* declined with increasing biomass, possibly indicating $N_2$ fixation, which requires lower number of heterocysts. Heterocyst formation is triggered by inorganic N depletion (e.g. Kumar et al., 2010) and in the Curonian Lagoon we observed the peak in heterocyst frequency (up to 15 per filament) in early summer as DIN was depleted to < 1 $\mu$M. Similar to other sites in the Baltic, the peak in heterocyst frequency was found when *A. flosaquae* biomass was still low (Walve and Larsson, 2007; Zakrisson and Larsson, 2014; Klawonn et al., 2016). Our estimates of *A. flosaquae* abundance were an order of magnitude higher than those previously reported for temperate and boreal

estuarine systems (Bentzon-Tilia et al., 2015; Klawonn et al., 2016; Olofsson et al., 2020a). These findings support the idea that *A. flosaquae* is the principal contributor to $N_2$ fixation in the brackish Baltic Sea and its adjacent coastal areas. The proliferation of heterocystous cyanobacteria in the Curonian Lagoon is favoured by P (Pilkaitytė and Razinkovas, 2007), which is released from sediments, particularly when bloom conditions result in high water column respiration and transient (night-time) depletion of oxygen (Zilius et al., 2014; Petkuviene et al., 2016). Moisander et al. (2007) demonstrated that P can enhance diazotrophic activity of heterocystous cyanobacteria in microcosms. Release of DIP from sediments may in turn enhance rates $N_2$ fixation resulting in a positive feedback for cyanobacteria bloom development. Measured summer DIP concentration (0.3 µM) in the Curonian Lagoon was similar to that in other Baltic coastal sites (e.g. Klawonn et al., 2016), suggesting that higher biomass might be also supported by higher N availability. A recent study by Broman et al. (submitted) suggests that $N_2$ fixation in the lagoon satisfies only 13% of N demand for phytoplankton. Thus, other internal sources such as N release from sediment and mineralization in the water column are important to meeting algal N demands.

The patchy distribution of cyanobacteria poses a significant challenge to reliably extrapolating results from site-specific measurements to the ecosystem scale (Zilius et al., 2014; 2018). Surface accumulation of positively-buoyant cyanobacteria and subsequent wind dispersion adds a dynamic component to biogeochemical processes in eutrophic lakes and estuarine systems (e.g. Gao et al., 2014; Zilius et al., 2014; Klawonn et al., 2015). Our previous work describing N fluxes in the Curonian Lagoon relied on a simple extrapolation of $N_2$ fixation rates measured at the two stations also used in this study. Here, we improve on our ability to scale up these measurements by using remote sensing of Chl-a to infer spatial and temporal variation in $N_2$ fixation. Our whole-lagoon estimates are based on data collected at stations within the northern and central portions of the lagoon, as access to the southern region is problematic. Hydrodynamic modeling studies have shown that water renewal times in the central and southern portions of the lagoon are comparable (Umgiesser et al., 2016). Monitoring data suggest that Chl-a and phytoplankton community composition is similar in the central and southern regions (Semenova and Dmitrieva, 2011; Bresciani et al. 2014; Vaičiūtė et al., 2021). Therefore, we felt it was appropriate to derive whole-lagoon estimates of N fixation based on *in situ* measurements from these two sites. We benefitted from prior work deriving Chl-a estimates from satellite images and their calibration to *in situ* measurements (Bresciani et al., 2014), but the success of the approach largely relied on the fact that heterocystous cyanobacteria dominated the summer–fall phytoplankton community of the lagoon, which provided a significant relationship between $N_2$ fixation and *in situ* Chl-a in surface layer. The regression model for estimating bottom layer $N_2$ fixation was marginally significant, and therefore we felt that the application of this model to deriving whole-water column rates was justified. Whole-lagoon estimates were not highly sensitive to assumed rates in the bottom layer because this layer accounts for a relative small proportion of the lagoon's volume and because measured $N_2$ fixation rates in the bottom layer were 7 times lower than the surface. However, it would be problematic to extrapolate this approach to periods outside of cyanobacteria dominance (e.g., spring diatom bloom) or to periods when other factors (e.g., low temperature in fall) constrain $N_2$ fixation. The transferability of this approach to other systems would likely depend on this facet; in systems where heterocystous cyanobacteria account for a small and variable fraction of Chl-a, the utility of Chl-a as a predictor of $N_2$ fixation

may be limited. Prior studies have used remote sensing to infer N₂ fixation, though by less direct means. For example, Hood et al. (2002) used SeaWiFS-derived estimates of *Trichodesmium* Chl-a, and modelled relationships between N₂ fixation and underwater irradiance to infer N₂ fixation in the tropical Atlantic Ocean. Coles et al. (2004) used remote sensing of Chl-a to estimate phytoplankton production in the North Atlantic and infer rates of N₂ fixation needed to support production. Other studies have related taxa-specific N₂ fixation to *in situ* measurements of Chl-a or algal biomass, including recent work in the Baltic Sea (Olofsson et al., 2020b). To our knowledge, ours is the first study to derive ecosystem-scale estimates by combining remote sensing of Chl-a with empirical models relating rates of N₂ fixation to Chl-a.

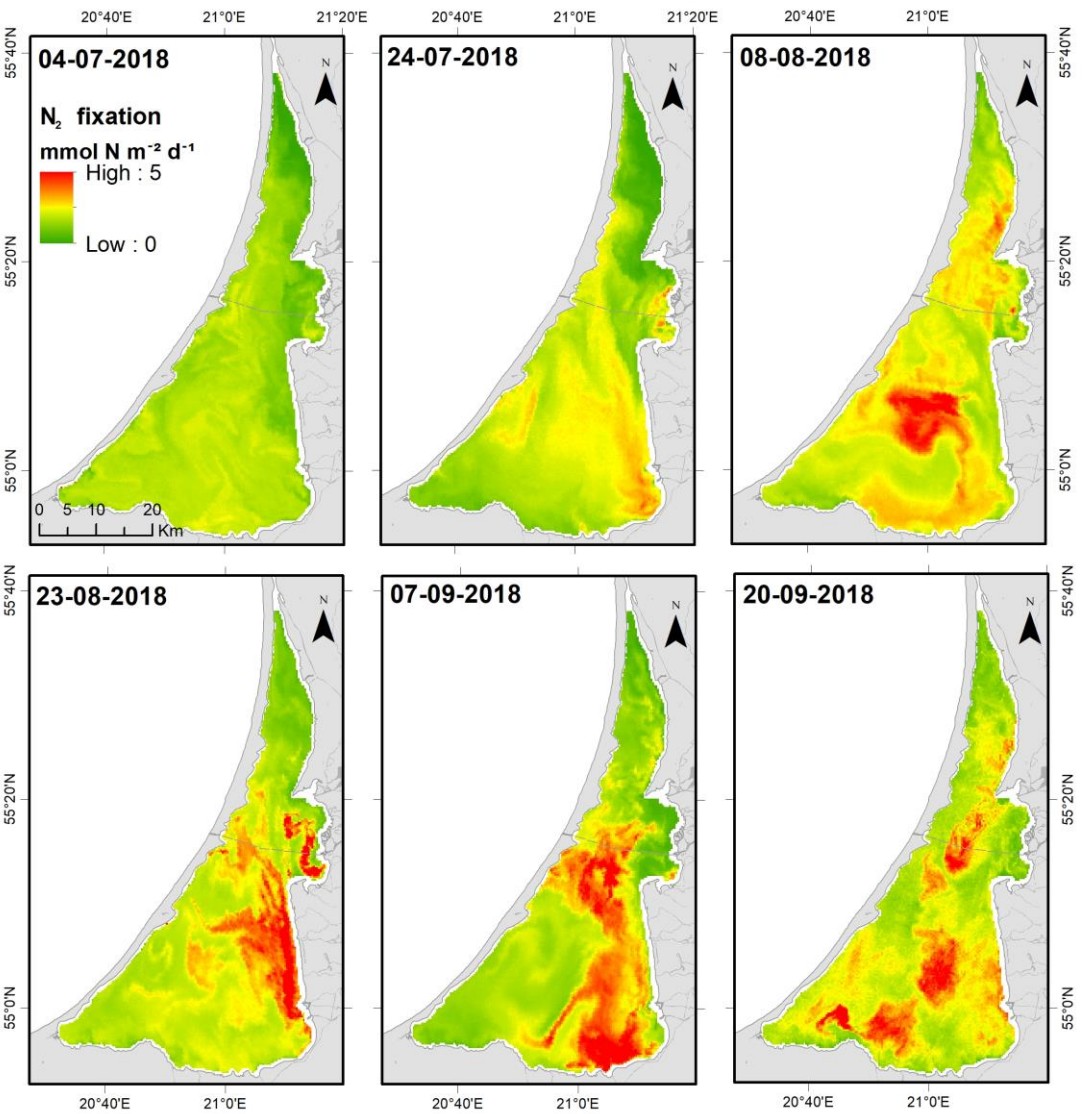

**Figure 8:** Estimates of pelagic N₂ fixation in the Curonian Lagoon derived from remote sensing of Chlorophyll-a.

Our remote sensing-based estimates of $N_2$ fixation for the Curonian Lagoon ranged from 1.5 to 2.5 mmol N m$^{-2}$ d$^{-1}$ (mean = 2.1 ± 0.1 mmol N m$^{-2}$ d$^{-1}$) during July–September. These estimates reveal that summer $N_2$ fixation rates are slightly lower in the Curonian Lagoon as compared to those measured at a coastal site of SW Baltic (3.6 ± 2.6 mmol N m$^{-2}$ d$^{-1}$; Klawonn et al., 2016), but higher than those found in the Great Belt (~ 1 mmol N m$^{-2}$ d$^{-1}$; Bentzon-Tilia et al., 2015), Baltic Proper (0.4 ± 0.1 mmol N m$^{-2}$ d$^{-1}$; Klawonn et al., 2016), and Bothnian Sea (0.6 ± 0.2 mmol N m$^{-2}$ d$^{-1}$; Olofsson et al., 2020b). By comparison, *in situ* estimates scaled using our prior method (based on the proportion of lagoon area represented by the two stations) yielded estimates ranging from 0.36 to 3.6 mmol N m$^{-2}$ d$^{-1}$ (mean = 1.9 ± 0.9 mmol N m$^{-2}$ d$^{-1}$) for corresponding dates. The two approaches yielded similar mean values, though with lower variability among those based on remote sensing. We attribute this to the shifting spatial distribution of cyanobacteria in the lagoon, which results in greater variability in site-specific measurements relative to the lagoon-scale assessments captured by remote sensing. These new estimates confirm our prior findings regarding the importance of $N_2$ fixation to the N balance of the lagoon. During periods of low river discharge, rates of $N_2$ fixation were twofold higher compared to monthly TN loads from the Nemunas River (June–October 2018 range = 0.83 to 1.1 mmol N m$^{-2}$ d$^{-1}$) (Supplement, Fig. S1). $N_2$ fixation during summer and fall largely offset annual average denitrification (3.2 mmol N m$^{-2}$ d$^{-1}$) and was equivalent to half of the measured sediment-water TDN exchange (3.8 mmol N m$^{-2}$ d$^{-1}$; Zilius et al., 2018). Our prior work also showed enhanced PN export to the Baltic Sea during periods when the lagoon was dominated by cyanobacteria. Positive buoyancy allows *A. flosaquae* and other cyanobacteria to remain suspended in the water column, which favours export in lagoon outflow, rather than retention via sedimentation (Bukaveckas et al., 2019). Overall, these findings suggest that the occurrence of heterocystous cyanobacteria blooms has substantially diminished the potential for the lagoon to attenuate N fluxes to the Baltic Sea. As blooms of $N_2$-fixing cyanobacteria are common among the three large Baltic lagoons (Curonian, Oder, and Vistula), their effect in diminishing lagoon N retention may be regionally important (Bangel et al., 2004; Dmitrieva and Semenova, 2012). Our approach using remote sensing combined with local, empirical models relating $N_2$ fixation to Chl-a may provide a useful means for assessing the role of cyanobacteria blooms in the context of N budgets for the Baltic Sea (e.g. Savchuk, 2018).

Our research has allowed us to better understand the environmental conditions that favour the occurrence of heterocystous cyanobacteria blooms and their contributions to the N budget of the lagoon. Important questions remain regarding the factors that regulate rates of $N_2$ fixation and the fate of atmospherically-derived N. Underwater irradiance is likely an important factor influencing biomass-specific $N_2$ fixation given its energetic costs. The Curonian Lagoon is a relatively turbid system in which the photic zone typically occupies less than 30% of the water column (Zilius et al., 2014). Our no-light incubations simulating the deeper layer of the south-central lagoon showed that $N_2$ fixation was occurring, but at biomass-specific rates that were 7-fold lower in comparison to the surface layer. Low $N_2$ fixation rates during dark incubations were also observed in cyanobacteria filaments collected from other coastal sites of the Baltic Sea (Svedén et al., 2015). Though heterocystous cyanobacteria continue to fix $N_2$ in the dark, it remains unknown for how long due to the high energetic costs. Previous measurements of $N_2$ and carbon fixation in *Aphanizomenon* from the Baltic Sea suggest that respiration of stored cell products

may provide energy for N$_2$ fixation under low light conditions (Svedén et al., 2015). Our study, as well as prior work, is based on 24-h incubations, simulating conditions at a fixed depth, which may not be indicative of rates that could be sustained by diazotrophs circulating over a range of depth and light conditions. Positive buoyancy and periodic mixing toward the surface may allow cyanobacteria to capture sufficient light energy to sustain N$_2$ fixation (Stal and Walsby, 2000). In addition to light availability, water temperature is likely an important constraint on seasonal patterns of N$_2$ fixation in temperate systems. Results from this, and a prior study (Zilius et al., 2018), show that despite a high abundance of *A. flosaquae* at the end of fall, heterocysts frequency, and thus N$_2$ fixation rates declined substantially when water temperature dropped below 15 °C. Zakrisson et al. (2014) suggested that temperature controls the enzymatic activity of nitrogenase, which directly regulates the intensity of N$_2$ fixation in filaments.

Recent work has shown that N fixed by diazotrophs is subsequently distributed to the planktonic food web (Woodland et al., 2013; Karlson et al., 2015), which likely involves a variety of mechanisms including grazing (Woodland et al., 2013), leakage of NH$_4^+$ and DON (Ohlendieck et al., 2007; Adam et al., 2016), and remineralization of N following algal senescence (Eglite et al., 2018). The relative importance of these pathways is not well known, though our data for the Curonian Lagoon suggests that heterotrophic bacteria, non-N fixing cyanobacteria and a diverse group of grazers benefit from the activities of heterocystous cyanobacteria. The biomass of heterotrophic bacteria increased during the bloom of heterocystous cyanobacteria to levels (250 µg C L$^{-1}$) that were appreciably higher than other coastal (Gulf of Finland and Archipelago Sea = 30–55 µg C L$^{-1}$) or open areas of the Baltic Sea (Bothnian Sea ~80 µg C L$^{-1}$; Baltic Proper = 16–44 µg C L$^{-1}$; Heinänen, 1991). It is likely that heterocystous cyanobacteria release dissolved organic matter which stimulates the growth of heterotrophic bacteria (Bertos-Fortis et al., 2016; Hoikkala et al., 2016; Berg et al., 2018; Berner et al., 2018). There is also evidence that non N$_2$-fixing cyanobacteria benefitted from the bloom of heterocystous cyanobacteria as indicated by higher abundance of *Microcystis* spp. and *Planktotrix agardhii*. Measured low δ$^{15}$N values (0.5 ± 0.2 ‰) in suspended living material suggest that fixed N can temporally support most of the nutritional needs for plankton (bacteria + phytoplankton) growth. Lastly, our prior work using stable isotopes tracked atmospherically-derived N from cyanobacteria to a diverse group of consumers and suggested that 50–80% of secondary production was supported by cyanobacteria during bloom events (Lesutiene et al., 2014).

In conclusion, our study contributes to a better understanding of the activity of coastal diazotrophs and their seasonal dynamics in eutrophic estuarine systems. The use of remote sensing allowed us to estimate N$_2$ fixation rates at the ecosystem scale and to show that these rates are high and relatively stable despite the dynamic and patchy distribution of cyanobacteria. The propensity for cyanobacteria to form dense, localized aggregates may influence the efficiency with which by-products of their carbon and N$_2$ fixation are disseminated by creating biogeochemical hotspots (Klawonn et al., 2015, 2019). Since intensifying blooms of cyanobacteria have already been observed in coastal areas of the Baltic Sea (Olofsson et al., 2020a), we may expect these blooms to have a stronger effect on ecosystem functioning in future. Therefore, further work combining

remote sensing and *in situ* studies may provide greater insights as to the fate of atmospherically-derived N and its implications for ecosystem energetics.

**Authors contribution**

MZ, PAB and DV conceived the ideas and designed methodology. MZ, IVL, SB and TB led the field survey and experimental activities. IV-L, DV, DO, EG, IL, SBr, AA and AZ assisted with analysis and data collection and analysis. AZ assisted with

420 statistical analyses. MZ and SB secured funding for the investigation. MV provided use of specialized facilities, MZ and PAB wrote the first draft of the paper, and all co-authors contributed to writing review and editing.

**Conflict of interest**

The authors declare that they have no conflict of interest.

**Acknowledgment**

We are in debt for the Coast Guard District of the State Border Guard Service for logistic support. We thank Jolita Petkuviene and Adele Mačiūtė for assistance in field sampling and laboratory analysis. We kindly thank the Editor and two anonymous reviewers for their constructive comments.

**Financial support**

This study has been supported by LMT grant "The role of atmospheric nitrogen fixation in the largest eutrophicated European

lagoon (NitFix)" (Agreement No. P-MIP-17-126).

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
