# Peer review of "Spatiotemporal patterns of $N_2$ fixation in coastal waters derived from rate measurements and remote sensing"

_Biogeosciences, 2020_

## Referee Comment (RC1) · Anonymous Referee #1 · 4 Dec 2020

The manuscript has an interesting dataset where the authors combine in situ measurement with satellite imaging to estimate areal nitrogen fixation with the benefit of reducing bias due to patchiness of cyanobacteria blooms. I have however a few concerns and questions to the authors to address. I therefore suggest a revision before considering it for publication.

Something that was surprising to me was how come you didn't find any picocyanobacteria? In Zilius et al. 2020 I interpret it as you had about 20% of the community during summer? Also in Klawonn et al. 2016, colonial picocyanobacterial comprise ca. 5-10% of the cyanobacterial community in terms of carbon. It seems like you sampled

on similar locations, maybe even at the same time, as in Zilius et al. 2020 so this needs an explanation. If it has to do with method differences, it needs to be explained or the statement of no picocyanobacterial removed and refer to previous study.

I am also a bit concerned about the method you use for measuring N2-fixtaion with injection of gas rather than pre-dissolved. I think this might cause an underestimation. Also the fact that you run 24 h incubations probably lead to underestimations of N2-fixation per h since they do less in the night when its dark (1.8 times less; Klawonn et al. 2016). I think a potential underestimation should be discussed and rates presented as per day since this is what you measure.

The theory of underestimation is further supported by that you have 1000-3000 ug cyanobacterial C per L and 120-200 nmol N2 fixation per h as compared to Klawonn 2016 where 100 ug cyanobacteria per L performed 80 nmol N2 fixation per h. Why do you think you have so low rates as compared to your biomass? Can it be P limitation?

What effects do you think the fact that cyanobacteria only comprised about up to 36 or 86% of the phytoplankton fraction has for your correlations with chlorophyll and further areal estimates of N2-fixation? I guess it must be very variable over the year how well your method can be applied? I think you should discuss this bias further.

More specific questions are listed below:

Line 33, comma after algae.

Line 39, I think you need to name this 2020a since it the first one appearing?

Line 45-46, when they are dead I guess? Maybe clarify that this is when they are detritus on the bottom.

Line 52, any references to the patchiness? Maybe Rolff et al. 2007?

Line 100, triplicates of samples or sampled from the same flask?

Line 109, maybe the Whatman and pore size should be on line 100 when first mentioned?

Line 120, I think it would be good to provide heterocysts per number of vegetative cells as well since they change in density over the season (Svedén et al. 2015).

Line 128, are not many picocyanobacterial smaller than 3 um? Is this what is commonly used for picocyanobacterial? Did you use any certain settings on the flow cytometer to determine picocyanobacteria, for example a cyano-specific filter? Did you use Sybr? Were they in the same sample as the heterotrophic bacteria or on its own? I am asking this since I am surprised that you did not see any, while you did in Zilius et al. 2020.

Line 133, does this mean that the flasks were top-filled without air during incubation? Did you shake/turn the flasks something to help with the mixing?

Line 137, in what way would pre-prepared isotopically enriched water be a risk of contamination? Contamination of what?

Line 139, I think its risk of underestimating rates when having flasks totally covered, 1% of light is still light and therefore it would have been more appropriate to have them covered instead. This would be good to mention in the results/discussion.

Line 148-150, did you measure the final labelled concentration in the flask or is this only an estimate from calculations? In case you only estimated the added concentration this can be a bias for your later rate calculations.

Line 153, how deep can you "see" with the satellites?

Line 181, why linear regression and not correlations? Don't you expect both of them to be interdependent rather than one dependent?

Line 189, please indicate in text and in figure legends when Chl is derived from in situ extractions and when from satellites.

Line 193, who are the non N2 fixing cyanobacteria if you did not have any pico-cyanobatceria? For example in November in Figure 2a?

Lines 201-202, this surprises me, no picocyanobacterial at all? Is this common here? In Zilius et al. 2020 you had at least about 20% of the biomass?

Line 205, among how many samples? Do you mean micrograms on the y-axes, please use a proper "micro" symbol for this.

Line 209, please explain the clustered numbers (months?) in the figure. Maybe also add a legend title including what the symbols are (biomass?).

Line 228, among how many samples?

Figure 5, I think it is better to show N2-fixation as per day since this is what you measured and since night-time is lower (See Klawonn et al. 2016 for night rates where they show the double rate at day time).

Figure 5b, is this low rates maybe related to lack of light?

Line 236, how does number of heterocysts relates to total biomass/number of vegetative cells of cyanobacteria?

Line 237, please abbreviate to A. flosaque throughout the manuscript except at first place mentioned.

Figure 6, do you mean "per ml" with per mil? How come there is so many heterocysts in November in the southern station but almost no N2 fixation nor cyanobacteria biomass at that time? In contrast the highest number of both N2 fixation and heterocysts numbers correlates for the northern station. This needs to be discussed. Also, It would be good to also have heterocysts per filaments/vegetative cells here to see how it changes over the season (Aphanizomenon heterocyst density varies with season; Svedén et al. 2015).

Line 248, I think you have light limited N2 fixation? 1% of surface light can still be 10-20 um photons, which can be sufficient for carbon fixation.

Lines 247-251, did you use correlations or regressions? If this is the regression models

you later use this must be clear.

Fig. 7 Clarify that the chl a from in situ extractions? Is this data from the whole year or only from the summer? For example if this is the whole year where is November value surface layer Chl of 250 ug L-1 but with no N2-fixation.

Line 259-260, can you provide the equation you used for these estimates?

Lines 260-261, how can you use this relationship when it was not significant, then there is no relationship?

Line 263, values of what?

Line 282, if you have ten times more Aphanizomenon you maybe should also have higher N2 fixation rates? This needs to be discussed and refer to data, for example Klawonn et al. 2016. Were they limited by something?

Line 298-300, but cyanobacteria did not dominate all the time and never close to 100%? How does this affect the results? When they are less then 50% of the Chl community is this not overestimating the N2-fixation rates? This needs to be discussed. The bottom layer had a lot of other organisms contributing to chl biomass.

Lines 309 and below, can you also put these areal N2-fixation estimates into perspective to other studies for the region? For example, Klawonn et al. 2016 and Olofsson et al. 2020 as well as references there in.

Line 343, how can the heterocyst frequency be so high without cyanobacteria biomass being high?

You need to discuss problems with N2 fixation from covered flasks and still standing flasks with gas injections somewhere in the discussion. "Caveats" with this study.

Line 500, Please change to Riemann.

---

## Author Comment (AC1) · 23 Dec 2020

The comment was uploaded in the form of a supplement:
https://bg.copernicus.org/preprints/bg-2020-419/bg-2020-419-AC1-supplement.pdf

**Response to Anonymous Referee #1**

**General comments**

The manuscript has an interesting dataset where the authors combine *in situ* measurement with satellite imaging to estimate areal nitrogen fixation with the benefit of reducing bias due to patchiness of cyanobacteria blooms. I have however a few concerns and questions to the authors to address. I therefore suggest a revision before considering it for publication.

Something that was surprising to me was how come you didn't find any picocyanobacteria? In Zilius et al. 2020 I interpret it as you had about 20% of the community during summer? Also in Klawonn et al. 2016, colonial picocyanobacterial comprise ca. 5-10% of the cyanobacterial community in terms of carbon. It seems like you sampled on similar locations, maybe even at the same time, as in Zilius et al. 2020 so this needs an explanation. If it has to do with method differences, it needs to be explained or the statement of no picocyanobacterial removed and refer to previous study.

**Answer:** We acknowledge the reviewer for their positive comments. In this study, taxa referred as "colonial picocyanobacteria" by the reviewer were found with microscopy counting, and due to their relatively low contribution (generally <2% of total biomass) they were assigned to "non-$N_2$-fixing cyanobacteria", and thus not further discussed in the submitted manuscript (Fig. 2). In the revised version of our manuscript, we have added information related to cyanobacteria composition and their biomass: "*Non-filamentous colonial cyanobacteria, such as Aphanocapsa spp., Aphanothece spp., Merismopedia spp. and Cyanodictyon spp. exhibited low biomass (< 2% of total) except in June, when their contribution reached 12% at the northern site (Fig. 2). Picocyanobacteria were not detected during the study period at either site.*" *(line 207-210)*

In Zilius et al. 2020, sequences were attributed to picocyanobacteria (not referring here as "colonial picocyanobacteria"). However, a volume of 50 to 70 ml was extracted for further sequencing and only few reads were assigned to picocyanobacteria. This means that picocyanobacteria were rare in this study and that they would not be detected by methods allowing quantification such as flow cytometry or epifluorescence microscopy. Both approaches are complementary and not contradictory since DNA methods can detect rare taxa but do not allow quantification yet.

I am also a bit concerned about the method you use for measuring $N_2$-fixation with injection of gas rather than pre-dissolved. I think this might cause an underestimation. Also the fact that you run 24 h incubations probably lead to underestimations of $N_2$-fixation per h since they do less in the night when its dark (1.8 times less; Klawonn et al. 2016). I think a potential underestimation should be discussed and rates presented as per day since this is what you measure.

**Answer:** Regarding the issue of hourly vs. daily rates of fixation, we agree with the reviewer's point that rates are likely to vary on a diel cycle (being lower at night). Therefore our diel incubations conducted under natural (outdoor) light conditions are more suitably expressed as daily rates than hourly rates since they are representative of both light and dark cycles. In the revised manuscript, we present daily values in figures and text.

With regards to methodology, we agree that there has been some debate about using the bubble method for $N_2$ fixation measurements (Mohr et al., 2010; Großkopf et al., 2012; White et al., 2020), but recent work (Wannicke et al., 2018) demonstrated that underestimation of rates is negligible (<1%) for incubations lasting 12–24 h. In the submitted version we have argued our choice for incubation duration: "*As the isotopic equilibration takes up to several hours (Mohr et al., 2010), we incubated the samples for 24 h, thus minimizing equilibration effects (Mulholland et al., 2012; Wannicke et al., 2018).*" *(line 136-138).* Eventually, our used technique avoids to have low labelling

**Fig. 1.** Responses to Reviewer comments

**Spatiotemporal patterns of N₂ fixation in coastal waters derived from rate measurements and remote sensing**

Mindaugas Zilius[1*], Irma Vybernaite-Lubiene[1], Diana Vaiciute[1], Donata Overlingė[1], Evelina Grinienė[1],
Anastasija Zaiko[2,3], Stefano Bonaglia[1,4], Iris Liskow[5], Maren Voss[5], Agneta Andersson[6], Sonia Brugel[6],
Tobia Politi[1], and Paul A. Bukaveckas[7*]

[1]Marine Research Institute, Klaipeda University, Klaipeda, 92294, Lithuania
[2]Coastal and Freshwater Group, Cawthron Institute, Nelson, 7042, New Zealand
[3]Zealand Institute of Marine Science, University of Auckland, Auckland, Private Bag 92019, New Zealand
[4]Department of Marine Sciences, University of Gothenburg, Box 461, Gothenburg, 40530, Sweden
[5]Department of Biological Oceanography, Leibniz Institute for Baltic Sea Research, Rostock, 18119, Germany
[6]Department of Ecology and Environmental Sciences, Umeå University, Umeå, 90187, Sweden
[7]Center for Environmental Studies, Virginia Commonwealth University, Richmond, VA 23284, USA

*Correspondence to*: Mindaugas Zilius (mindaugas.zilius@jmtc.ku.lt) and Paul A. Bukaveckas (pabukaveckas@vcu.edu)

**Abstract.** Coastal lagoons are important sites for nitrogen (N) removal via sediment burial and denitrification. Blooms of heterocystous cyanobacteria may diminish N retention as dinitrogen (N₂) fixation offsets atmospheric losses via denitrification. We measured N₂ fixation in the Curonian Lagoon, Europe's largest coastal lagoon, to better understand the factors controlling N₂ fixation in the context of seasonal changes in phytoplankton community composition and external N inputs. Temporal patterns in N₂ fixation were primarily determined by the abundance of heterocystous cyanobacteria, mainly *Aphanizomenon flosaquae*, which became abundant after the decline in riverine nitrate inputs associated with snowmelt. Heterocystous cyanobacteria dominated the summer phytoplankton community resulting in strong correlations between chlorophyll-a (Chl-a) and N₂ fixation. We used regression models relating N₂ fixation to Chl-a, along with remote sensing-based estimates of Chl-a to derive lagoon-scale estimates of N₂ fixation. N₂ fixation by pelagic cyanobacteria was found to be a significant component of the lagoon's N budget based on comparisons to previously derived fluxes associated with riverine inputs, sediment-water exchange and losses via denitrification. To our knowledge, this is the first study to derive ecosystem-scale estimates of N₂ fixation by combining remote sensing of Chl-a with empirical models relating N₂ fixation rates to Chl-a.

**Fig. 2.** Revised manuscript

**Supplement:**

**Response to Anonymous Referee #1**

**General comments**

The manuscript has an interesting dataset where the authors combine *in situ* measurement with satellite imaging to estimate areal nitrogen fixation with the benefit of reducing bias due to patchiness of cyanobacteria blooms. I have however a few concerns and questions to the authors to address. I therefore suggest a revision before considering it for publication.

Something that was surprising to me was how come you didn't find any picocyanobacteria? In Zilius et al. 2020 I interpret it as you had about 20% of the community during summer? Also in Klawonn et al. 2016, colonial picocyanobacterial comprise ca. 5-10% of the cyanobacterial community in terms of carbon. It seems like you sampled on similar locations, maybe even at the same time, as in Zilius et al. 2020 so this needs an explanation. If it has to do with method differences, it needs to be explained or the statement of no picocyanobacterial removed and refer to previous study.

**Answer:** We acknowledge the reviewer for their positive comments. In this study, taxa referred as "colonial picocyanobacteria" by the reviewer were found with microscopy counting, and due to their relatively low contribution (generally <2% of total biomass) they were assigned to "non-N2-fixing cyanobacteria", and thus not further discussed in the submitted manuscript (Fig. 2). In the revised version of our manuscript, we have added information related to cyanobacteria composition and their biomass: "*Non-filamentous colonial cyanobacteria, such as Aphanocapsa spp., Aphanothece spp., Merismopedia spp. and Cyanodictyon spp. exhibited low biomass (< 2% of total) except in June, when their contribution reached 12% at the northern site (Fig. 2). Picocyanobacteria were not detected during the study period at either site." (line 207-210)*

In Zilius et al. 2020, sequences were attributed to picocyanobacteria (not referring here as "colonial picocyanobacteria"). However, a volume of 50 to 70 ml was extracted for further sequencing and only few reads were assigned to picocyanobacteria. This means that picocyanobacteria were rare in this study and that they would not be detected by methods allowing quantification such as flow cytometry or epifluorescence microscopy. Both approaches are complementary and not contradictory since DNA methods can detect rare taxa but do not allow quantification yet.

I am also a bit concerned about the method you use for measuring  $N_2$ -fixation with injection of gas rather than pre-dissolved. I think this might cause an underestimation. Also the fact that you run 24 h incubations probably lead to underestimations of  $N_2$ -fixation per h since they do less in the night when its dark (1.8 times less; Klawonn et al. 2016). I think a potential underestimation should be discussed and rates presented as per day since this is what you measure.

**Answer:** Regarding the issue of hourly vs. daily rates of fixation, we agree with the reviewer's point that rates are likely to vary on a diel cycle (being lower at night). Therefore our diel incubations conducted under natural (outdoor) light conditions are more suitably expressed as daily rates than hourly rates since they are representative of both light and dark cycles. In the revised manuscript, we present daily values in figures and text.

With regards to methodology, we agree that there has been some debate about using the bubble method for N2 fixation measurements (Mohr et al., 2010; Großkopf et al., 2012; White et al., 2020), but recent work (Wannicke et al., 2018) demonstrated that underestimation of rates is negligible (<1%) for incubations lasting 12–24 h. In the submitted version we have argued our choice for incubation duration: "As the isotopic equilibration takes up to several hours (Mohr et al., 2010), we incubated the samples for 24 h, thus minimizing equilibration effects (Mulholland et al., 2012; Wannicke et al., 2018." (line 136-138). Eventually, our used technique avoids to have low labelling

(percentage label should be between 5-10%) as the labelled seawater method often results in low quantities of  ${}^{15}N_2$  gas in the water (e.g. Klawonn et al. (2015) had only 1% label in their experiment).

The theory of underestimation is further supported by that you have 1000-3000  $\mu$ g cyanobacterial C per L and 120-200 nmol N2 fixation per h as compared to Klawonn 2016 where 100  $\mu$ g cyanobacteria per L performed 80 nmol N2 fixation per h. Why do you think you have so low rates as compared to your biomass? Can it be P limitation?

**Answer:** We agree with the reviewer's point that P limitation may play a role in limiting N fixation in the Curonian Lagoon. In a prior study, it was shown that P additions stimulated growth rates of  $N_2$  fixing cyanobacteria from the Curonian Lagoon (Pilkaitytė and Razinkovas, 2007). Likewise, addition of P stimulated diazotrophic community resulting in elevated  $N_2$  fixation rates (Moisander et al. 2007). We may expect that dissolved P was limiting, which constrained  $N_2$  fixation during summer . Thus, we suppose that DIP release from sediment and higher biomass of *Aphanizomenon* and diazotrophic activity frequently observed in the end of summer (Zilius et al. 2014, 2018) are not coincidence but rather consequence of increased P availability. We have modified the Discussion to address this point "*The proliferation of heterocystous cyanobacteria in the Curonian Lagoon is favoured by P* (*Pilkaityté and Razinkovas, 2007*), which is released from sediments, particularly when bloom conditions result in high water column respiration and transient (night-time) depletion of oxygen (Petkuviene et al., 2016; Zilius et al., 2014). Moisander et al. (2007) demonstrated that P can enhance diazotrophic activity of heterocystous cyanobacteria in microcosms. Release of dissolved P from sediments may in turn enhance rates  $N_2$  fixation resulting in a positive feedback for cyanobacteria bloom development." (line 299-304)

What effects do you think the fact that cyanobacteria only comprised about up to 36 or 86% of the phytoplankton fraction has for your correlations with chlorophyll and further areal estimates of  $N_2$ -fixation? I guess it must be very variable over the year how well your method can be applied? I think you should discuss this bias further.

**Answer:** With regard to our ability to model  $N_2$  fixation on the basis of Ch-a, we specifically address this point in the discussion: "We benefitted from prior work deriving Chl-a estimates from satellite images and their calibration to in situ measurements (Bresciani et al., 2014), but the success of the approach largely relied on the fact that heterocystous cyanobacteria dominated the summer–fall phytoplankton community of the lagoon, which provided a significant correlation between  $N_2$  fixation and in situ Chl-a. However, it would be problematic to extrapolate this approach to periods outside of cyanobacteria dominance (e.g., spring diatom bloom) or to periods when other factors (e.g., low temperature in fall) constrain  $N_2$  fixation." (line 312-317)

**References**

Großkopf, T., Mohr, W., Baustian, T., Schunck, H., Gill, D., Kuypers, M.M.M., Lavik, G., Schmitz, R.A., Wallace, D.W.R., LaRoche, J.: Doubling of marine dinitrogen-fixation rates based on direct measurements. Nature, 488, 361–364, 2012

Moisander, P. H., W. Paerl, H. W., Dyble, J., and Sivonen, K.: Phosphorus limitation and diel control of nitrogen-fixing cyanobacteria in the Baltic Sea. Mar. Ecol. Prog. Ser., 345, 41–50, doi: 10.3354/meps06964, 2007.

Mulholland, M. R., Bernhardt, P. W., Blanco-Garcia, J. L., Mannino, A., Hyde, K., Mondragon, E., Turk, K., Moisander, P. H., and Zehr, J. P.: Rates of dinitrogen fixation and the abundance of diazotrophs in North American coastal waters between Cape Hatteras and Georges Bank. Limnol. Oceanogr., 57, 1067–1083, doi:10.4319/lo.2012.57.4.1067, 2012.

Pilkaitytė, R., and Razinkovas, A.: Seasonal changes in phytoplankton composition and nutrient limitation in a shallow Baltic lagoon. Boreal Environ. Res., 12, 551–559, 2007.

Wannicke, N., Benavides, M., Dalsgaard, T., Dippner, J. W., Montoya, J. P., and Voss, M.: New perspectives on nitrogen fixation measurements using  $^{15}\mathrm{N}_2$  gas. Front. Mar. Sci., 5, 120, doi:10.3389/fmars.2018.00120, 2018.

White, A.E, Granger, J., Selden, C., Gradoville, M.R., Potts, L., Bourbonnais, A., Fulweiler, R.W., Knapp, A., Mohr, W., Moisander, P.H., Tobias, C.R., Caffin, M., Wilson, S.T., Benavides, M., Bonnet, S., Mulholland, M.R., and Chang, X.B.: A critical review of the 15N2 tracer method to measure diazotrophic production in pelagic ecosystems. Limnol. Oceanogr.-Meth. doi:10.1002/lom3.10353

**Specific comments**

Line 45-46, when they are dead I guess? Maybe clarify that this is when they are detritus on the bottom.

**Answer**: we assume that respiration of living cells rather detritus cause bottom hypoxia. During summer blooms, when plankton (mainly cyanobacteria and heterotrophic bacteria) respiration exceed diffusive oxygen supply to deeper layer, benthic community eventually depletes oxygen from adjacent bottom. We have clarified sentence and it reads now "*Large blooms of living cyanobacteria are associated with high oxygen demand in the water column, which results in transient (night-time) bottom hypoxia and enhances the release of dissolved inorganic phosphorus (DIP) from sediments (Petkuviene et al., 2016; Zilius et al., 2014)." (line 45-47)*

Line 52, any references to the patchiness? Maybe Rolff et al. 2007?

Answer: thanks for suggestion. We have included this reference.

Line 100, triplicates of samples or sampled from the same flask?

**Answer**: we collected water samples in triplicates, and in the laboratory each of them were filtered separately. Text modified: "*Triplicate water samples from each site or layer were filtered (Whatman GF/F, pore size 0.7 \mum) for inorganic and organic nutrient analysis as previously described by Vybernaite-Lubiene et al. (2017)." (line 100-101)*

Line 109, maybe the Whatman and pore size should be on line 100 when first mentioned?

Answer: corrected accordingly.

Line 120, I think it would be good to provide heterocysts per number of vegetative cells as well since they change in density over the season (Svedén et al. 2015).

**Answer**: following the reviewer's suggestion, we added estimates of heterocyst frequency per cyanobacteria filament in the revised version of the manuscript, see updated Figure 6:

**Fig. 6.** Abundance of heterocysts of *Dolichospermum* spp. and *A. flosaquae* (a, b), heterocyst frequency per filament (c, d) and stable isotope composition of particulate nitrogen ( $\delta^{15}$ N-PN) at southern and northern sites in the Curonian Lagoon during 2018.  $\delta^{15}$ N-PN values are mean and standard error (some error bars not visible).

In the text, we have added following information:

"The number of heterocysts (cell  $L^{-1}$ ) and their frequency per millimeter of filament (mm-1) was also determined." (line 121-122)

"Total heterocyst frequency per filament was higher at the beginning of summer (up to  $15 \text{ mm}^{-1}$ ) at both sites, and gradually declined afterwards (Fig. 6c, d)." (line 249-250)

Line 128, are not many picocyanobacterial smaller than 3 um? Is this what is commonly used for picocyanobacterial? Did you use any certain settings on the flow cytometer to determine picocyanobacteria, for example a cyano-specific filter? Did you use Sybr? Were they in the same sample as the heterotrophic bacteria or on its own? I am asking this since I am surprised that you did not see any, while you did in Zilius et al. 2020.

**Answer**: we kindly note the use of the term "colonial picocyanobacteria" is misleading, since "colonial picocyanobacteria" do not belong to picoplankton owing to their colony size, and need to be enumerated by methods designed for nano- and microplankton. "Colonial picocyanobacteria" refer to cyanobacteria cells of 1-3 µm size embedded in mucilaginous colonies. The colonies are most commonly over 10 µm size and not very abundant (less than 1 colony mL-1) in the Baltic Sea. As the colonies are large and mucilaginous inverted microscopy after sedimentation is the preferred method for detection and quantification. Following HELCOM recommendations, the biomass is estimated by converting in biovolume-carbon each cell from the colony. These colonies have typically small abundances <1 colony mL-1. A larger volume of sample is required to detect such cyanobacterial taxa. In Klawonn et al (2016), free-living picocyanobacteria have not been counted, though they are present and abundant in Baltic Proper waters (B1 or BY31 stations). They counted the "colonial taxa.

picocyanobacteria" by inverted microscopy after sedimentation of 25 ml of Lugol-preserved samples. In this study, 10 to 25 ml of Lugol-preserved samples were counted by inverted microscopy and the cyanobacteria with cells <3  $\mu$ m in colonies were assigned to the "non-N2-fixing cyanobacteria" category. *Aphanocapsa* spp., *Aphanothece* spp., *Merismopedia* spp. and *Cyanodictyon* spp. were detected in low biomass (< 2% of total phytoplankton biomass) during the study period at either site, except in June when their contribution reached 12% at the northern site, as it is now specified in the text "Non-filamentous colonial cyanobacteria, such as Aphanocapsa spp., Aphanothece spp., *Merismopedia spp. and Cyanodictyon spp. exhibited low biomass (< 2% of total) except in June, when their contribution reached 12% at the northern site (Fig. 2). Picocyanobacteria were not detected during the study period at either site." (line 207-210)*

Picocyanobacteria refer to free-living unicellular cyanobacteria with a size below 2 or 3  $\mu$ m depending on the size definition chosen. And we used this original definition in the manuscript. They are free-living, belong to the picoplankton and are usually abundant (over 100 cells mL-1) when present. Then can be detected and/or counted by flow cytometry or epifluorescence microscopy, methods designed to count picoplankton. By flow cytometry they are typically counted in volumes of 50-100  $\mu$ L, as flow cytometry is designed to count small and abundant cells/particles.

In the present study, picocyanobacteria were counted with a flow cytometer following standard procedures. The preservation procedure, the running settings (flow rate, acquisition time, etc.) followed standard recommendations. The analyses for picocyanobacteria were performed independently from the analyses for heterotrophic bacteria. The BD Accuri C6 allows the detection of fluorescence from phycoerythrin (at 585/40 nm after excitation at 488 nm), phycocyanin (at 675/25 nm after excitation at 640 nm) and chlorophyll (>670 nm after excitation at 488 nm). During the analyses many cells showing chlorophyll fluorescence were detected but with no higher phycoerythrin or phycocyanin fluorescence over background level. Therefore, we concluded that no picocyanobacteria was detected in this study.

Line 133, does this mean that the flasks were top-filled without air during incubation? Did you shake/turn the flasks something to help with the mixing?

**Answer**: yes, the bottle was completely filled, and after injection of gas bubble was gently mixed. The missing information was added: "*The samples were filled into 500 ml transparent HDPE bottles and carefully sealed preventing formation of air bubbles. Each sample received 0.5 ml*15N2 (98% 15N2, Sigma-Aldrich) injected by syringe through a gas-tight septum, and then gently mixed for 10 min (*Zilius et al., 2018*)." (line 134-136)

Line 137, in what way would pre-prepared isotopically enriched water be a risk of contamination? Contamination of what?

**Answer:** we mean that all 15N label have been excreted into the surrounding waters within the incubation time is likely immediately reused and thus appears on the filters. During short-time incubations, this is in particularly relevant when proportion of excreted 15N is relative close to quantities of dissolved 15N2 gas, which can happen when using labelled seawater method water with low tracer percentage. To avoid any confusion this statement was removed from the text.

Line 139, I think its risk of underestimating rates when having flasks totally covered, 1% of light is still light and therefore it would have been more appropriate to have them covered instead. This would be good to mention in the results/discussion.

**Answer**: we disagree with this point. Measured PAR at 2 m depth was always < 5  $\mu$ mol m-2 s-1 (June–November), which is well below 1% of surface water irradiance, see added information in revised version "Surface water samples were incubated outdoors at ambient irradiance, while samples from 2.0–3.5 m were wrapped in aluminium foil as in situ irradiance was below 1% of surface

*PAR at these depths* (< 5  $\mu$ mol m-2 s-1 *in the period of June–November*)." (line 138-140). We appreciate that near-dark may not be quite the same as dark, but given the very low rates relative to surface (photic) values, we feel that this would not appreciably affect our findings. We feel that the more important methodological issue is that in these studies samples are almost always incubated at a fixed light intensity, whereas cyanobacteria mixing in the water column experience a dynamic light environment. This point is made in the Material and Methods: *"However, such fixed dark conditions is less representative to in situ conditions as cyanobacteria colonies can migrate upward to surface photic zone or use limited light for photosynthesis."* (line 140-142) and in the Discussion: *"Our study, as well as prior work, is based on 24-h incubations, simulating conditions at a fixed depth, which may not be indicative of rates that could be sustained by diazotrophs circulating over a range of depth and light conditions."* (line 363-365)

Line 148-150, did you measure the final labelled concentration in the flask or is this only an estimate from calculations? In case you only estimated the added concentration this can be a bias for your later rate calculations.

**Answer:** unfortunately, 15N2 concentration was not quantified in bottles. Therefore, it may lead underestimation of rates as suggested by White et al. 2020. Though we are aware of method use, there are number of studies still published without testing 15N2 concentration in incubated bottles. This information was added in the revised version of manuscript: "As we have used theoretical estimation of 15N2 gas dissolution in bottles instead quantification with membrane inlet mass spectrometer, it can result in some underestimation of rates (White et al., 2020)." (line 153-154)

**References**

White, A.E, Granger, J., Selden, C., Gradoville, M.R., Potts, L., Bourbonnais, A., Fulweiler, R.W., Knapp, A., Mohr, W., Moisander, P.H., Tobias, C.R., Caffin, M., Wilson, S.T., Benavides, M., Bonnet, S., Mulholland, M.R., Chang, X.B. 2020. A critical review of the  $^{15}N_2$  tracer method to measure diazotrophic production in pelagic ecosystems. Limnology and Oceanography: Methods. doi:10.1002/lom3.10353

Line 153, how deep can you "see" with the satellites?

**Answer:** Optical remote sensing, i.e. the method based on passive radiometers operating in the visible and near-infrared wavelengths, is the only one which penetrates the surface of the waterbody (Robinson, 2010). The satellites can observe the water down to one optical depth, the portion of the water column where approximately 90% of the remote sensing observed signal originates (Gordon and McCluney, 1975; Werdell and Bailey, 2005). The optical depth is equivalent to the inverse of the diffuse attenuation coefficient (Kd) (Gordon and McCluney, 1975) and has also been shown to empirically relate to the Secchi disk depth (Lee et al., 2018). The range of Kd(490) in Swedish coastal waters of the Baltic Sea during 2008 was  $0.31-1.19 \text{ m}^{-1}$  (Kratzer, Vinterhav, 2010). In Curonian Lagoon, estimated Kd value from daily buoy measurements was  $2.7-5.7 \text{ m}^{-1}$  during presence of cyanobacteria (2014-2015). This information was added in Material and Methods, see lines 159-163.

**References**

Gordon, H.R., McCluney, W.R., 1975. Estimation of the depth of sunlight penetration in the sea for remote sensing. Appl. Opt. 14 (2), 413–416.

Lee, Z., Shang, S., Du, K., Wei, J., 2018. Resolving the long-standing puzzles about the observed Secchi depth relationships. Limnol. Oceanogr. https://doi.org/10.1002/lno.10940.

Kratzer, S., Vinterhav, Ch., 2010. Improvement of MERIS level 2 products in Baltic Sea coastal areas by applying the Improved Contrast between Ocean and Land processor (ICOL) – data analysis and validation. Oceanologia 52 (2), 211–236. DOI: 10.5697/oc.52-2.211

Robinson, I., 2010. Discovering the Ocean from Space. The unique applications of satellite oceanography. Springer-Verlag Berlin Heidelberg, p. 638.

Werdell, P.J., Bailey, S.W., 2005. An improved in-situ bio-optical data set for ocean color algorithm development and satellite data product validation. Remote Sens. Environ. 98 (1), 122–140.

Line 181, why linear regression and not correlations? Don't you expect both of them to be interdependent rather than one dependent?

**Answer:** while we consider that correlation coefficient represents the direction and strength of the relationship between chlorophyll and  $N_2$  fixation rates, the regression coefficient determines the effect of chlorophyll a (independent variable) on the  $N_2$  fixation (dependent variable), and determine the explained variation.

Line 189, please indicate in text and in figure legends when Chl is derived from *in situ* extractions and when from satellites.

**Answer:** thanks for the suggestion.

Line 193, who are the non  $N_2$  fixing cyanobacteria if you did not have any picocyanobatceria? For example in November in Figure 2a?

**Answer**: the main non-N2-fxing cyanobacteria are represented by *Planktotrix agardhii* and *Microcystis* spp. This information was also added in the text: "*The non-N2-fixing cyanobacteria were dominated by Planktotrix agardhii and Microcystis spp.*" (line 203-204)

Lines 201-202, this surprises me, no picocyanobacterial at all? Is this common here? In Zilius et al. 2020 you had at least about 20% of the biomass?

Answer: see detail answer above.

Line 205, among how many samples? Do you mean micrograms on the y-axes, please use a proper "micro" symbol for this.

**Answer:** in the revised version, we have added information that chlorophyll a are presented as mean values and standard error (some error bars not visible) based on three replicates. Yes, on y-axis all units are in micrograms, according to the suggestions we applied proper style of "micro".

Line 209, please explain the clustered numbers (months?) in the figure. Maybe also add a legend title including what the symbols are (biomass?).

**Answer**: the information was added, and it is "*Phytoplankton biomass and community composition* were generally similar between surface and bottom layers (April–September), except in October– November when the abundance of N2-fixing cyanobacteria was greater in the surface layer (2500– 3500 µg C L-1) relative to the bottom layer (<100 µg C L-1)." (line 212-214). Also we have updated legend in Figure 2.

Line 228, among how many samples?

**Answer**: data are represented by mean values and standard error based on 3 replicates. Missing information was added to figure captions "*Figure 4*: *Temporal patterns in temperature, dissolved organic carbon (a, b), dissolved and organic nitrogen (c, e), phosphorus (e, f), and DIN:DIP ratios (g, h) at southern (left panel) and northern (right panel) sites in the Curonian Lagoon during 2018 (error bars denote standard error based on 3 replicates; some not visible).*"

Figure 5, I think it is better to show  $N_2$ -fixation as per day since this is what you measured and since night-time is lower (See Klawonn et al. 2016 for night rates where they show the double rate at day time).

**Answer**: Thanks for the suggestion. In revised version, volumetric N2-fixation rates are presented per day ( $\mu$ mol L-1 d-1).

Figure 5b, is this low rates maybe related to lack of light?

**Answer**: yes, these rates were measured in the dark.

Line 236, how does number of heterocysts relates to total biomass/number of vegetative cells of cyanobacteria?

**Answer**: We have plotted heterocyst abundance versus number of vegetative cells of N2-fixing cyanobacteria, which indeed provided nice results: "*The abundance of heterocysts varied seasonally depending on the number of vegetative cells of N*2-fixing cyanobacteria (y=0.0251x+75.0,  $R^2$ =0.92) with lowest values less than 2000 cells  $L^{-1}$  and peak values exceeding 2 million cells  $L^{-1}$  in late summer." (line 246-248)

Line 237, please abbreviate to A. flosaque throughout the manuscript except at first place mentioned.

Answer: done.

Figure 6, do you mean "per ml" with per mil? How come there is so many heterocysts in November in the southern station but almost no  $N_2$  fixation nor cyanobacteria biomass at that time? In contrast the highest number of both  $N_2$  fixation and heterocysts numbers correlates for the northern station. This needs to be discussed. Also, It would be good to also have heterocysts per filaments/vegetative cells here to see how it changes over the season (*Aphanizomenon* heterocyst density varies with season; Svedén et al. 2015).

**Answer**: "per mil" means delta units that are expressed in molecules per thousand, but for convenience we have change to "%".

In revised version, we also provided heterocyst frequency, which better corresponded to N2 fixation dynamic. The updated Figure 6 shows that patterns in heterocyst frequency was relatively low in November coinciding to decreased N2 fixation rates. We assume that the October-November period represents the decline of the cyanobacteria bloom. In the submitted version we discussed that "*Results from this, and a prior study (Zilius et al. 2018), show that despite a high abundance of A. flosaquae at the end of fall, heterocyst frequency, and thus N2 fixation rates declined substantially when water temperature dropped below 15 °C. Zakrisson et al. (2014) suggested that temperature controls the enzymatic activity of nitrogenase, which directly regulates the intensity of N2 fixation in filaments." (line 363-367)*

Line 248, I think you have light limited N2 fixation? 1% of surface light can still be 10-20  $\mu$ m photons, which can be sufficient for carbon fixation.

**Answer**: we agree that  $N_2$  fixation can be light limited in the turbid Curonian lagoon, but we note that the photic zone does not extend into the bottom layer (2–3.5 m depth), therefore we feel it was appropriate to incubate the bottom samples in the dark (see response to prior comment).

Lines 247-251, did you use correlations or regressions? If this is the regression models you later use this must be clear.

**Answer**: in present study, we have used linear regression, which later allowed to derived N2 fixation estimates based on remote sensing Chl-a. We have reformulated the sentence to avoid confusion, and now it reads " $N_2$  fixation in the surface layer was significantly (p < 0.001) predicted by in situ

Chl-a concentration ( $R^2 = 0.91$ ), A. flosaquae biomass ( $R^2 = 0.83$ ) and A. flosaquae heterocysts ( $R^2 = 0.88$  all p < 0.001; Fig. 7). Whereas N2 fixation in the bottom layer was weakly explained by in situ Chl-a ( $R^2 = 0.52$ , p = 0.07), but not A. flosaquae biomass or heterocysts. In situ chlorophyll-specific N2 fixation derived from regression equations (Fig. 7a, b) was considerably lower in the deeper layer (0.002 ± 0.001 µmol N µg-1 Chl-a d-1) relative to the surface layer (0.018 ± 0.002 µmol N µg-1 Chl-a d-1)." (line 258-262)

Fig. 7 Clarify that the ChI a from in situ extractions? Is this data from the whole year or only from the summer? For example if this is the whole year where is November value surface layer ChI of 250  $\mu$ g L-1 but with no N2-fixation.

**Answer:** we agree that some information is lacking. Not it reads "*Figure 7: Relationships between*  $N_2$  fixation and in situ measured Chlorophyll-a (a, b), Aphanizomenon flosaquae biomass (c, d) and their heterocysts (e, f) in surface (northern and southern sites) and bottom (southern site) layers of the Curonian Lagoon during April–September 2018." (line 264-266)

Line 259-260, can you provide the equation you used for these estimates?

**Answer**: equation for this estimation is already showed in Fig. 7a, but to be clear we have added this equation in brackets: "We used the relationship between  $N_2$  fixation and Chl-a (y = 0.018x - 0.459; Fig. 7a) for the surface layer to derive estimates of  $N_2$  fixation for the upper water column (0– 2 m)." (line 270-271)

Lines 260-261, how can you use this relationship when it was not significant, then there is no relationship?

**Answer**: The regression is marginally significant (p = 0.08) and has a reasonable R2 value (0.52). Therefore, we felt that this model provided the best means for estimating bottom layer N2 fixation. We also note that the bottom layer accounts for a relative small proportion of the lagoon's volume and an even smaller proportion of N2 fixation (since surface rates are 7x higher). Therefore, our whole-lagoon estimates of N2 fixation are not highly sensitive to assumptions about bottom water rates.

Line 263, values of what?

**Answer**: it was referred to estimated N2 fixation rates. Now sentence reads "The impact of the bloom on N2 fixation can be visualized from the relatively low and uniform estimated rates throughout the lagoon during July, and the subsequent development of localized hotspots in the southern lagoon during August and September (Fig. 8)." (line 275-276)

Line 282, if you have ten times more *Aphanizomenon* you maybe should also have higher  $N_2$  fixation rates? This needs to be discussed and refer to data, for example Klawonn et al. 2016. Were they limited by something?

Answer: see our earlier answer.

Line 298-300, but cyanobacteria did not dominate all the time and never close to 100%? How does this affect the results? When they are less then 50% of the Chl community is this not overestimating the  $N_2$ -fixation rates? This needs to be discussed. The bottom layer had a lot of other organisms contributing to chl biomass.

Answer: see our earlier answer.

Lines 309 and below, can you also put these areal  $N_2$ -fixation estimates into perspective to other studies for the region? For example, Klawonn et al. 2016 and Olofsson et al. 2020 as well as references there in.

**Answer:** thanks for suggestion. We have put our estimates in the context of the Baltic region: "*These* estimates reveal that summer  $N_2$  fixation rates are slightly lower in the coastal site of SW Baltic (3.6  $\pm$  2.6 µmol  $m^{-2}$  d-1; Klawonn et al., 2016), but higher than those found in the Great Belt (~ 1 mmol  $m^{-2}$  d-1; Bentzon-Tilia et al., 2015), Baltic Proper (0.4  $\pm$  0.1 mmol  $m^{-2}$  d-1; Klawonn et al., 2016), and Bothnian Sea (0.6  $\pm$  0.2 mol  $m^{-2}$  d-1; Olofsson et al. 2020b)." (line 328-331)

Line 343, how can the heterocyst frequency be so high without cyanobacteria biomass being high?

**Answer:** Heterocyst abundance tracks cyanobacteria abundance, but in the revised version of manuscript, we also show that heterocyst frequency per filament decreases when biomass increases in November (see figure above).

You need to discuss problems with  $N_2$  fixation from covered flasks and still standing flasks with gas injections somewhere in the discussion. "Caveats" with this study.

**Answer**: in the revised manuscript, we have stated that "Our study, as well as prior work, is based on 24-h incubations, simulating conditions at a fixed depth, which may not be indicative of rates that could be sustained by diazotrophs circulating over a range of depth and light conditions." (line 359-361)

**Technical corrections**

Line 33, comma after algae.

Answer: added.

Line 39, I think you need to name this 2020a since it the first one appearing?

Answer: corrected.

Line 500, Please change to Riemann

Answer: done.

---

## Referee Comment (RC3) · Anonymous Referee #1 · 4 Jan 2021

**Response to Author comment #1 (provided in blue)**

I am pleased to see that the authors have improved the manuscript according to the comments below. Which includes being open about some caveats, comparison to other studies, some methods that were missing and additional data on heterocyst frequency. However, there is still some minor adjustments I would like to suggest before moving on. See specific comments below.

**Reviewer:** The manuscript has an interesting dataset where the authors combine *in situ* measurement with satellite imaging to estimate areal nitrogen fixation with the benefit of reducing bias due to patchiness of cyanobacteria blooms. I have however a few concerns and questions to the authors to address. I therefore suggest a revision before considering it for publication. Something that was surprising to me was how come you didn't find any picocyanobacteria? In Zilius et al. 2020 I interpret it as you had about 20% of the community during summer? Also in Klawonn et al. 2016, colonial picocyanobacterial comprise ca. 5-10% of the cyanobacterial community in terms of carbon. It seems like you sampled on similar locations, maybe even at the same time, as in Zilius et al. 2020 so this needs an explanation. If it has to do with method differences, it needs to be explained or the statement of no picocyanobacterial removed and refer to previous study.

**Answer:** We acknowledge the reviewer for their positive comments. In this study, taxa referred as "colonial picocyanobacteria" by the reviewer were found with microscopy counting, and due to their relatively low contribution (generally <2% of total biomass) they were assigned to "non-$N_2$-fixing cyanobacteria", and thus not further discussed in the submitted manuscript (Fig. 2). In the revised version of our manuscript, we have added information related to cyanobacteria composition and their biomass: "*Non-filamentous colonial cyanobacteria, such as Aphanocapsa spp., Aphanothece spp., Merismopedia spp. and Cyanodictyon spp. exhibited low biomass (< 2% of total) except in June, when their contribution reached 12% at the northern site (Fig. 2). Picocyanobacteria were not detected during the study period at either site.*" (line 207-210)

In Zilius et al. 2020, sequences were attributed to picocyanobacteria (not referring here as "colonial picocyanobacteria"). However, a volume of 50 to 70 ml was extracted for further sequencing and only few reads were assigned to picocyanobacteria. This means that picocyanobacteria were rare in this study and that they would not be detected by methods allowing quantification such as flow cytometry or epifluorescence microscopy. Both approaches are complementary and not contradictory since DNA methods can detect rare taxa but do not allow quantification yet.

Reviewer 1: Thank you for clarifying this in the revision of the manuscript and for looking further into this by also applying microscopy in addition to flow cytometry. Feel free to also include this extra information so that future readers do not confuse between the groups of colonial vs. free-living picocyanobacteria.

I am also a bit concerned about the method you use for measuring $N_2$-fixation with injection of gas rather than pre-dissolved. I think this might cause an underestimation. Also the fact that you run 24 h incubations probably lead to underestimations of $N_2$-fixation per h since they do less in the night when its dark (1.8 times less; Klawonn et al. 2016). I think a potential underestimation should be discussed and rates presented as per day since this is what you measure.

**Answer:** Regarding the issue of hourly vs. daily rates of fixation, we agree with the reviewer's point that rates are likely to vary on a diel cycle (being lower at night). Therefore our diel incubations conducted under natural (outdoor) light conditions are more suitably expressed as daily rates than hourly rates since they are representative of both light and dark cycles. In the revised manuscript, we present daily values in figures and text.

With regards to methodology, we agree that there has been some debate about using the bubble method for $N_2$ fixation measurements (Mohr et al., 2010; Großkopf et al., 2012; White et al., 2020), but recent work (Wannicke et al., 2018) demonstrated that underestimation of rates is

negligible (<1%) for incubations lasting 12–24 h. In the submitted version we have argued our choice for incubation duration: "*As the isotopic equilibration takes up to several hours (Mohr et al., 2010), we incubated the samples for 24 h, thus minimizing equilibration effects (Mulholland et al., 2012; Wannicke et al., 2018.*" *(line 136-138).* Eventually, our used technique avoids to have low labelling (percentage label should be between 5-10%) as the labelled seawater method often results in low quantities of $^{15}N_2$ gas in the water (e.g. Klawonn et al. (2015) had only 1% label in their experiment).

Reviewer 1: I am not fully sure if the last sentence about the optimum labelling ranges is within the revision or only a reply to the comment. But if it is in the text, do you have a reference for the mentioned optimum labelling percentages? I do not see any problem with having 1 % labelling when using the dilution approach (which you refer to in Klawonn et al.) since you then have this labelling in the whole flask already from the beginning, and as long as you can trace it and measure the final concentration. In case there is no suitable reference for the optimum ranges, and you want to mention it in the manuscript, I would suggest that you rephrase it to only mention that this range of percentages works well when using the bubble method and not compare to Klawonn et al. as an example of where it has not worked well(?) since I do not know if you have proof of this, and its slightly different methods. Further, I think it is great that you changed to 24 h values since this is what you measured and it includes both day and night.

The theory of underestimation is further supported by that you have 1000-3000 µg cyanobacterial C per L and 120-200 nmol N2 fixation per h as compared to Klawonn 2016 where 100 µg cyanobacteria per L performed 80 nmol $N_2$ fixation per h. Why do you think you have so low rates as compared to your biomass? Can it be P limitation?

**Answer:** We agree with the reviewer's point that P limitation may play a role in limiting N fixation in the Curonian Lagoon. In a prior study, it was shown that P additions stimulated growth rates of $N_2$ fixing cyanobacteria from the Curonian Lagoon (Pilkaitytė and Razinkovas, 2007*)*. Likewise, addition of P stimulated diazotrophic community resulting in elevated $N_2$ fixation rates (Moisander et al. 2007). We may expect that dissolved P was limiting, which constrained $N_2$ fixation during summer. Thus, we suppose that DIP release from sediment and higher biomass of *Aphanizomenon* and diazotrophic activity frequently observed in the end of summer (Zilius et al. 2014, 2018) are not coincidence but rather consequence of increased P availability. We have modified the Discussion to address this point "*The proliferation of heterocystous cyanobacteria in the Curonian Lagoon is favoured by P (Pilkaitytė and Razinkovas, 2007), which is released from sediments, particularly when bloom conditions result in high water column respiration and transient (night-time) depletion of oxygen (Petkuviene et al., 2016; Zilius et al., 2014). Moisander et al. (2007) demonstrated that P can enhance diazotrophic activity of heterocystous cyanobacteria in microcosms. Release of dissolved P from sediments may in turn enhance rates $N_2$ fixation resulting in a positive feedback for cyanobacteria bloom development.*" (line 299-304)

Reviewer 1: In addition to stating that they are favoured by it you should maybe also say that this might be why they are performing compare-wise lower N2 fixation in comparison to other regions of the Baltic Sea and provide some reference to P concentrations in the lagoon during summer. Maybe they also have enough of other sources of N, such as ammonium, to support some of their needs?

What effects do you think the fact that cyanobacteria only comprised about up to 36 or 86% of the phytoplankton fraction has for your correlations with chlorophyll and further areal estimates of $N_2$-fixation? I guess it must be very variable over the year how well your method can be applied? I think you should discuss this bias further.

**Answer:** With regard to our ability to model $N_2$ fixation on the basis of Ch-a, we specifically address this point in the discussion: "*We benefitted from prior work deriving Chl-a estimates from satellite images and their calibration to in situ measurements (Bresciani et al., 2014), but the success of the*

*approach largely relied on the fact that heterocystous cyanobacteria dominated the summer–fall phytoplankton community of the lagoon, which provided a significant correlation between $N_2$ fixation and in situ Chl-a. However, it would be problematic to extrapolate this approach to periods outside of cyanobacteria dominance (e.g., spring diatom bloom) or to periods when other factors (e.g., low temperature in fall) constrain $N_2$ fixation.*" (line 312-317)

References

Großkopf, T., Mohr, W., Baustian, T., Schunck, H., Gill, D., Kuypers, M.M.M., Lavik, G., Schmitz, R.A., Wallace, D.W.R., LaRoche, J.: Doubling of marine dinitrogen-fixation rates based on direct measurements. Nature, 488, 361–364, 2012

Moisander, P. H., W. Paerl, H. W., Dyble, J., and Sivonen, K.: Phosphorus limitation and diel control of nitrogen-fixing cyanobacteria in the Baltic Sea. Mar. Ecol. Prog. Ser., 345, 41–50, doi: 10.3354/meps06964, 2007.

Mulholland, M. R., Bernhardt, P. W., Blanco-Garcia, J. L., Mannino, A., Hyde, K., Mondragon, E., Turk, K., Moisander, P. H., and Zehr, J. P.: Rates of dinitrogen fixation and the abundance of diazotrophs in North American coastal waters between Cape Hatteras and Georges Bank. Limnol. Oceanogr., 57, 1067–1083, doi:10.4319/lo.2012.57.4.1067, 2012.

Pilkaitytė, R., and Razinkovas, A.: Seasonal changes in phytoplankton composition and nutrient limitation in a shallow Baltic lagoon. Boreal Environ. Res., 12, 551–559, 2007.

Wannicke, N., Benavides, M., Dalsgaard, T., Dippner, J. W., Montoya, J. P., and Voss, M.: New perspectives on nitrogen fixation measurements using $^{15}N_2$ gas. Front. Mar. Sci., 5, 120, doi:10.3389/fmars.2018.00120, 2018.

White, A.E, Granger, J., Selden, C., Gradoville, M.R., Potts, L., Bourbonnais, A., Fulweiler, R.W., Knapp, A., Mohr, W., Moisander, P.H., Tobias, C.R., Caffin, M., Wilson, S.T., Benavides, M., Bonnet, S., Mulholland, M.R., and Chang, X.B.: A critical review of the $^{15}N_2$ tracer method to measure diazotrophic production in pelagic ecosystems. Limnol. Oceanogr.-Meth. doi:10.1002/lom3.10353

**Specific comments**

Line 45-46, when they are dead I guess? Maybe clarify that this is when they are detritus on the bottom.

**Answer**: we assume that respiration of living cells rather detritus cause bottom hypoxia. During summer blooms, when plankton (mainly cyanobacteria and heterotrophic bacteria) respiration exceed diffusive oxygen supply to deeper layer, benthic community eventually depletes oxygen from adjacent bottom. We have clarified sentence and it reads now "*Large blooms of living cyanobacteria are associated with high oxygen demand in the water column, which results in transient (night-time) bottom hypoxia and enhances the release of dissolved inorganic phosphorus (DIP) from sediments (Petkuviene et al., 2016; Zilius et al., 2014).*" (line 45-47)

Reviewer 1: Thank you for the explanation. Although this might be part of the problem, I think a majority of the oxygen on the bottom is consumed as they die and need to be degraded. I think you should add a sentence and a reference to this as well if you mentioned the above statement. See for example Conley et al. 2009 on Hypoxia in the Baltic Sea.

Line 52, any references to the patchiness? Maybe Rolff et al. 2007?

**Answer**: thanks for suggestion. We have included this reference.

Line 100, triplicates of samples or sampled from the same flask?

**Answer**: we collected water samples in triplicates, and in the laboratory each of them were filtered

separately. Text modified: "*Triplicate water samples from each site or layer were filtered (Whatman GF/F, pore size 0.7 μm) for inorganic and organic nutrient analysis as previously described by Vybernaite-Lubiene et al. (2017).*" (line 100-101)

Line 109, maybe the Whatman and pore size should be on line 100 when first mentioned?

**Answer**: corrected accordingly.

Line 120, I think it would be good to provide heterocysts per number of vegetative cells as well since they change in density over the season (Svedén et al. 2015).

**Answer**: following the reviewer's suggestion, we added estimates of heterocyst frequency per cyanobacteria filament in the revised version of the manuscript, see updated Figure 6:

[Figure]

**Fig. 6.** Abundance of heterocysts of *Dolichospermum* spp. and *A. flosaquae* (a, b), heterocyst frequency per filament (c, d) and stable isotope composition of particulate nitrogen (δ15N-PN) at southern and northern sites in the Curonian Lagoon during 2018. δ15N-PN values are mean and standard error (some error bars not visible).

In the text, we have added following information:

„*The number of heterocysts (cell L$^{-1}$) and their frequency per millimeter of filament (mm$^{-1}$) was also determined.*" (line 121-122)

„*Total heterocyst frequency per filament was higher at the beginning of summer (up to 15 mm$^{-1}$) at both sites, and gradually declined afterwards (Fig. 6c, d).*" (line 249-250)

Reviewer 1: Thank you for including heterocysts per filaments, I think this is more informative than just heterocysts per ml. However, I think you should be more specific. In the southern site the numbers were highest in June and August/September, so actually two peaks, and in the northern sites they peaked in June/July and August depending on the species, so not only early in the summer

Line 128, are not many picocyanobacterial smaller than 3 um? Is this what is commonly used for picocyanobacterial? Did you use any certain settings on the flow cytometer to determine picocyanobacteria, for example a cyano-specific filter? Did you use Sybr? Were they in the same sample as the heterotrophic bacteria or on its own? I am asking this since I am surprised that you did not see any, while you did in Zilius et al. 2020.

**Answer**: we kindly note the use of the term "colonial picocyanobacteria" is misleading, since "colonial picocyanobacteria" do not belong to picoplankton owing to their colony size, and need to be enumerated by methods designed for nano- and microplankton. "Colonial picocyanobacteria" refer to cyanobacteria cells of 1-3 µm size embedded in mucilaginous colonies. The colonies are most commonly over 10 µm size and not very abundant (less than 1 colony mL$^{-1}$) in the Baltic Sea. As the colonies are large and mucilaginous inverted microscopy after sedimentation is the preferred method for detection and quantification. Following HELCOM recommendations, the biomass is estimated by converting in biovolume-carbon each cell from the colony. These colonies have typically small abundances <1 colony mL$^{-1}$. A larger volume of sample is required to detect such cyanobacterial taxa. In Klawonn et al (2016), free-living picocyanobacteria have not been counted, though they are present and abundant in Baltic Proper waters (B1 or BY31 stations). They counted the "colonial picocyanobacteria" by inverted microscopy after sedimentation of 25 ml of Lugol-preserved samples. In this study, 10 to 25 ml of Lugol-preserved samples were counted by inverted microscopy and the cyanobacteria with cells <3 µm in colonies were assigned to the "non-N$_2$-fixing cyanobacteria" category. *Aphanocapsa* spp., *Aphanothece* spp., *Merismopedia* spp. and *Cyanodictyon* spp. were detected in low biomass (< 2% of total phytoplankton biomass) during the study period at either site, except in June when their contribution reached 12% at the northern site, as it is now specified in the text "*Non-filamentous colonial cyanobacteria, such as Aphanocapsa spp., Aphanothece spp., Merismopedia spp. and Cyanodictyon spp. exhibited low biomass (< 2% of total) except in June, when their contribution reached 12% at the northern site (Fig. 2). Picocyanobacteria were not detected during the study period at either site.*" (line 207-210)

Picocyanobacteria refer to free-living unicellular cyanobacteria with a size below 2 or 3 µm depending on the size definition chosen. And we used this original definition in the manuscript. They are free-living, belong to the picoplankton and are usually abundant (over 100 cells mL$^{-1}$) when present. Then can be detected and/or counted by flow cytometry or epifluorescence microscopy, methods designed to count picoplankton. By flow cytometry they are typically counted in volumes of 50-100 µL, as flow cytometry is designed to count small and abundant cells/particles.

In the present study, picocyanobacteria were counted with a flow cytometer following standard procedures. The preservation procedure, the running settings (flow rate, acquisition time, etc.) followed standard recommendations. The analyses for picocyanobacteria were performed independently from the analyses for heterotrophic bacteria. The BD Accuri C6 allows the detection of fluorescence from phycoerythrin (at 585/40 nm after excitation at 488 nm), phycocyanin (at 675/25 nm after excitation at 640 nm) and chlorophyll (>670 nm after excitation at 488 nm). During the analyses many cells showing chlorophyll fluorescence were detected but with no higher phycoerythrin or phycocyanin fluorescence over background level. Therefore, we concluded that no picocyanobacteria was detected in this study.

Reviewer 1: Thank you for your explanation and clarification.

Line 133, does this mean that the flasks were top-filled without air during incubation? Did you shake/turn the flasks something to help with the mixing?

**Answer**: yes, the bottle was completely filled, and after injection of gas bubble was gently mixed. The missing information was added: "*The samples were filled into 500 ml transparent HDPE bottles and carefully sealed preventing formation of air bubbles. Each sample received 0.5 ml $^{15}N_2$ (98%*

*[15]N2, Sigma-Aldrich) injected by syringe through a gas-tight septum, and then gently mixed for 10 min (Zilius et al., 2018).”* (line 134-136)

Reviewer 1: Good that you added the gently mixing, I think this is important to know when reading the methods.

Line 137, in what way would pre-prepared isotopically enriched water be a risk of contamination? Contamination of what?

**Answer:** we mean that all $^{15}N$ label have been excreted into the surrounding waters within the incubation time is likely immediately reused and thus appears on the filters. During short-time incubations, this is in particularly relevant when proportion of excreted $^{15}N$ is relative close to quantities of dissolved $^{15}N_2$ gas, which can happen when using labelled seawater method water with low tracer percentage. To avoid any confusion this statement was removed from the text.

Line 139, I think its risk of underestimating rates when having flasks totally covered, 1% of light is still light and therefore it would have been more appropriate to have them covered instead. This would be good to mention in the results/discussion.

**Answer**: we disagree with this point. Measured PAR at 2 m depth was always < 5 µmol m$^{-2}$ s$^{-1}$ (June–November), which is well below 1% of surface water irradiance, see added information in revised version "*Surface water samples were incubated outdoors at ambient irradiance, while samples from 2.0–3.5 m were wrapped in aluminium foil as in situ irradiance was below 1% of surface PAR at these depths (< 5 µmol m$^{-2}$ s$^{-1}$ in the period of June–November).*" (line 138-140). We appreciate that near-dark may not be quite the same as dark, but given the very low rates relative to surface (photic) values, we feel that this would not appreciably affect our findings. We feel that the more important methodological issue is that in these studies samples are almost always incubated at a fixed light intensity, whereas cyanobacteria mixing in the water column experience a dynamic light environment. This point is made in the Material and Methods: "*However, such fixed dark conditions is less representative to in situ conditions as cyanobacteria colonies can migrate upward to surface photic zone or use limited light for photosynthesis.*" (line 140-142) and in the Discussion: "*Our study, as well as prior work, is based on 24-h incubations, simulating conditions at a fixed depth, which may not be indicative of rates that could be sustained by diazotrophs circulating over a range of depth and light conditions.*" (line 363-365)

Reviewer 1: Thank you for the clarification, I did not understand that the light was that low, good that you included the light level measured in the water.

Line 148-150, did you measure the final labelled concentration in the flask or is this only an estimate from calculations? In case you only estimated the added concentration this can be a bias for your later rate calculations.

**Answer:** unfortunately, $^{15}N_2$ concentration was not quantified in bottles. Therefore, it may lead underestimation of rates as suggested by White et al. 2020. Though we are aware of method use, there are number of studies still published without testing $^{15}N_2$ concentration in incubated bottles. This information was added in the revised version of manuscript: "*As we have used theoretical estimation of $^{15}N_2$ gas dissolution in bottles instead quantification with membrane inlet mass spectrometer, it can result in some underestimation of rates (White et al., 2020).*" (line 153-154)

Reviewer 1: I think it is good that you included this as a potential caveat.

References
White, A.E, Granger, J., Selden, C., Gradoville, M.R., Potts, L., Bourbonnais, A., Fulweiler, R.W., Knapp, A., Mohr, W., Moisander, P.H., Tobias, C.R., Caffin, M., Wilson, S.T., Benavides, M., Bonnet,

S., Mulholland, M.R., Chang, X.B. 2020. A critical review of the $^{15}N_2$ tracer method to measure diazotrophic production in pelagic ecosystems. Limnology and Oceanography: Methods. doi:10.1002/lom3.10353

Line 153, how deep can you "see" with the satellites?

**Answer:** Optical remote sensing, i.e. the method based on passive radiometers operating in the visible and near-infrared wavelengths, is the only one which penetrates the surface of the waterbody (Robinson, 2010). The satellites can observe the water down to one optical depth, the portion of the water column where approximately 90% of the remote sensing observed signal originates (Gordon and McCluney, 1975; Werdell and Bailey, 2005). The optical depth is equivalent to the inverse of the diffuse attenuation coefficient ($K_d$) (Gordon and McCluney, 1975) and has also been shown to empirically relate to the Secchi disk depth (Lee et al., 2018). The range of $K_d(490)$ in Swedish coastal waters of the Baltic Sea during 2008 was 0.31–1.19 $m^{-1}$ (Kratzer, Vinterhav, 2010). In Curonian Lagoon, estimated $K_d$ value from daily buoy measurements was 2.7–5.7 $m^{-1}$ during presence of cyanobacteria (2014-2015). This information was added in Material and Methods, see lines 159-163.

Reviewer 1: Thank you for the explanation.

References
Gordon, H.R., McCluney, W.R., 1975. Estimation of the depth of sunlight penetration in the sea for remote sensing. Appl. Opt. 14 (2), 413–416.
Lee, Z., Shang, S., Du, K., Wei, J., 2018. Resolving the long-standing puzzles about the observed Secchi depth relationships. Limnol. Oceanogr. https://doi.org/10.1002/lno.10940.
Kratzer, S., Vinterhav, Ch., 2010. Improvement of MERIS level 2 products in Baltic Sea coastal areas by applying the Improved Contrast between Ocean and Land processor (ICOL) – data analysis and validation. Oceanologia 52 (2), 211–236. DOI: 10.5697/oc.52-2.211

Robinson, I., 2010. Discovering the Ocean from Space. The unique applications of satellite oceanography. Springer-Verlag Berlin Heidelberg, p. 638.
Werdell, P.J., Bailey, S.W., 2005. An improved in-situ bio-optical data set for ocean color algorithm development and satellite data product validation. Remote Sens. Environ. 98 (1), 122–140.

Line 181, why linear regression and not correlations? Don't you expect both of them to be interdependent rather than one dependent?

**Answer:** while we consider that correlation coefficient represents the direction and strength of the relationship between chlorophyll and $N_2$ fixation rates, the regression coefficient determines the effect of chlorophyll a (independent variable) on the $N_2$ fixation (dependent variable), and determine the explained variation.

Reviewer 1: I guess this is fine as long as you state this as your tested hypothesis in the methods.

Line 189, please indicate in text and in figure legends when Chl is derived from *in situ* extractions and when from satellites.

**Answer:** thanks for the suggestion.

Line 193, who are the non $N_2$ fixing cyanobacteria if you did not have any picocyanobatceria? For example in November in Figure 2a?

**Answer**: the main non-$N_2$-fxing cyanobacteria are represented by *Planktotrix agardhii* and *Microcystis* spp. This information was also added in the text: "*The non-$N_2$-fixing cyanobacteria were dominated by Planktotrix agardhii and Microcystis spp.*" (line 203-204)

Lines 201-202, this surprises me, no picocyanobacterial at all? Is this common here? In Zilius et al. 2020 you had at least about 20% of the biomass?

**Answer**: see detail answer above.

Line 205, among how many samples? Do you mean micrograms on the y-axes, please use a proper "micro" symbol for this.

**Answer:** in the revised version, we have added information that chlorophyll a are presented as mean values and standard error (some error bars not visible) based on three replicates. Yes, on y-axis all units are in micrograms, according to the suggestions we applied proper style of "micro".

Line 209, please explain the clustered numbers (months?) in the figure. Maybe also add a legend title including what the symbols are (biomass?).

**Answer**: the information was added, and it is "*Phytoplankton biomass and community composition were generally similar between surface and bottom layers (April–September), except in October– November when the abundance of $N_2$-fixing cyanobacteria was greater in the surface layer (2500– 3500 µg C $L^{-1}$) relative to the bottom layer (<100 µg C $L^{-1}$).*" (line 212-214). Also we have updated legend in Figure 2.

Line 228, among how many samples?

**Answer**: data are represented by mean values and standard error based on 3 replicates. Missing information was added to figure captions "***Figure 4***: *Temporal patterns in temperature, dissolved organic carbon (a, b), dissolved and organic nitrogen (c, e), phosphorus (e, f), and DIN:DIP ratios (g, h) at southern (left panel) and northern (right panel) sites in the Curonian Lagoon during 2018 (error bars denote standard error based on 3 replicates; some not visible).*"

Figure 5, I think it is better to show $N_2$-fixation as per day since this is what you measured and since night-time is lower (See Klawonn et al. 2016 for night rates where they show the double rate at day time).

**Answer**: Thanks for the suggestion. In revised version, volumetric $N_2$-fixation rates are presented per day ($µmol\ L^{-1}\ d^{-1}$).

Figure 5b, is this low rates maybe related to lack of light?

**Answer**: yes, these rates were measured in the dark.

Line 236, how does number of heterocysts relates to total biomass/number of vegetative cells of cyanobacteria?

**Answer**: We have plotted heterocyst abundance versus number of vegetative cells of $N_2$-fixing cyanobacteria, which indeed provided nice results: "*The abundance of heterocysts varied seasonally depending on the number of vegetative cells of $N_2$-fixing cyanobacteria (y=0.0251x+75.0, $R^2$=0.92) with lowest values less than 2000 cells $L^{-1}$ and peak values exceeding 2 million cells $L^{-1}$ in late summer.*" (line 246-248)

Line 237, please abbreviate to *A. flosaque* throughout the manuscript except at first place mentioned.

**Answer**: done.

Figure 6, do you mean "per ml" with per mil? How come there is so many heterocysts in November in the southern station but almost no $N_2$ fixation nor cyanobacteria biomass at that time? In contrast

the highest number of both $N_2$ fixation and heterocysts numbers correlates for the northern station. This needs to be discussed. Also, It would be good to also have heterocysts per filaments/vegetative cells here to see how it changes over the season (*Aphanizomenon* heterocyst density varies with season; Svedén et al. 2015).

**Answer**: "per mil" means delta units that are expressed in molecules per thousand, but for convenience we have change to "‰".

In revised version, we also provided heterocyst frequency, which better corresponded to $N_2$ fixation dynamic. The updated Figure 6 shows that patterns in heterocyst frequency was relatively low in November coinciding to decreased $N_2$ fixation rates. We assume that the October-November period represents the decline of the cyanobacteria bloom. In the submitted version we discussed that "*Results from this, and a prior study (Zilius et al. 2018), show that despite a high abundance of A. flosaquae at the end of fall, heterocyst frequency, and thus $N_2$ fixation rates declined substantially when water temperature dropped below 15 °C. Zakrisson et al. (2014) suggested that temperature controls the enzymatic activity of nitrogenase, which directly regulates the intensity of $N_2$ fixation in filaments.*" (line 363-367)

Reviewer 1: I am still a bit perplexed about how chl a of cyanobacteria I November at the southern site is very low, the heterocysts per filaments is very low but the heterocysts per L is very high? I understand that N2 fixation goes down with temperature, but how can it be so many heterocysts when there is no biomass?

Line 248, I think you have light limited $N_2$ fixation? 1% of surface light can still be 10-20 µm photons, which can be sufficient for carbon fixation.

**Answer**: we agree that $N_2$ fixation can be light limited in the turbid Curonian lagoon, but we note that the photic zone does not extend into the bottom layer (2–3.5 m depth), therefore we feel it was appropriate to incubate the bottom samples in the dark (see response to prior comment).

Lines 247-251, did you use correlations or regressions? If this is the regression models you later use this must be clear.

**Answer**: in present study, we have used linear regression, which later allowed to derived $N_2$ fixation estimates based on remote sensing Chl-a. We have reformulated the sentence to avoid confusion, and now it reads "*$N_2$ fixation in the surface layer was significantly (p < 0.001) predicted by in situ Chl-a concentration ($R^2$ = 0.91), A. flosaquae biomass ($R^2$ = 0.83) and A. flosaquae heterocysts ($R^2$ = 0.88 all p < 0.001; Fig. 7). Whereas $N_2$ fixation in the bottom layer was weakly explained by in situ Chl-a ($R^2$ = 0.52, p =0.07), but not A. flosaquae biomass or heterocysts. In situ chlorophyll-specific $N_2$ fixation derived from regression equations (Fig. 7a, b) was considerably lower in the deeper layer (0.002 ± 0.001 µmol N µg$^{-1}$ Chl-a d$^{-1}$) relative to the surface layer (0.018 ± 0.002 µmol N µg$^{-1}$ Chl-a d$^{-1}$).*" (line 258-262)

Fig. 7 Clarify that the Chl a from in situ extractions? Is this data from the whole year or only from the summer? For example if this is the whole year where is November value surface layer Chl of 250 µg L$^{-1}$ but with no $N_2$-fixation.

**Answer:** we agree that some information is lacking. Not it reads "*Figure 7: Relationships between $N_2$ fixation and in situ measured Chlorophyll-a (a, b), Aphanizomenon flosaquae biomass (c, d) and their heterocysts (e, f) in surface (northern and southern sites) and bottom (southern site) layers of the Curonian Lagoon during April–September 2018.*" (line 264-266)

Line 259-260, can you provide the equation you used for these estimates?

**Answer**: equation for this estimation is already showed in Fig. 7a, but to be clear we have added this equation in brackets: "*We used the relationship between $N_2$ fixation and Chl-a (y = 0.018x – 0.459; Fig. 7a) for the surface layer to derive estimates of $N_2$ fixation for the upper water column (0–2 m).*" (line 270-271)

Lines 260-261, how can you use this relationship when it was not significant, then there is no relationship?

**Answer**: The regression is marginally significant ($p = 0.08$) and has a reasonable $R^2$ value (0.52). Therefore, we felt that this model provided the best means for estimating bottom layer $N_2$ fixation. We also note that the bottom layer accounts for a relative small proportion of the lagoon's volume and an even smaller proportion of $N_2$ fixation (since surface rates are 7x higher). Therefore, our whole-lagoon estimates of $N_2$ fixation are not highly sensitive to assumptions about bottom water rates.

Reviewer 1: Is this caveat explained also in the manuscript?

Line 263, values of what?

**Answer**: it was referred to estimated $N_2$ fixation rates. Now sentence reads "*The impact of the bloom on $N_2$ fixation can be visualized from the relatively low and uniform estimated rates throughout the lagoon during July, and the subsequent development of localized hotspots in the southern lagoon during August and September (Fig. 8).*" (line 275-276)

Line 282, if you have ten times more *Aphanizomenon* you maybe should also have higher $N_2$ fixation rates? This needs to be discussed and refer to data, for example Klawonn et al. 2016. Were they limited by something?

**Answer**: see our earlier answer.

Line 298-300, but cyanobacteria did not dominate all the time and never close to 100%? How does this affect the results? When they are less then 50% of the Chl community is this not overestimating the $N_2$-fixation rates? This needs to be discussed. The bottom layer had a lot of other organisms contributing to chl biomass.

**Answer:** see our earlier answer.

Lines 309 and below, can you also put these areal $N_2$-fixation estimates into perspective to other studies for the region? For example, Klawonn et al. 2016 and Olofsson et al. 2020 as well as references there in.

**Answer:** thanks for suggestion. We have put our estimates in the context of the Baltic region: "*These estimates reveal that summer $N_2$ fixation rates are slightly lower in the coastal site of SW Baltic (3.6 ± 2.6 µmol m$^{-2}$ d$^{-1}$; Klawonn et al., 2016), but higher than those found in the Great Belt (~ 1 mmol m$^{-2}$ d$^{-1}$; Bentzon-Tilia et al., 2015), Baltic Proper (0.4 ± 0.1 mmol m$^{-2}$ d$^{-1}$; Klawonn et al., 2016), and Bothnian Sea (0.6 ± 0.2 mol m$^{-2}$ d$^{-1}$; Olofsson et al. 2020b).*" (line 328-331)

Reviewer 1. Please use the same units across all studies so its easier for the reader to compare. Maybe you need to formulate the sentence a bit clearer: "These estimates reveal that summer N2 fixation rates are slightly lower in the Curonian Lagoon as compared to those measured at a coastal site of…"

Line 343, how can the heterocyst frequency be so high without cyanobacteria biomass being high?

**Answer:** Heterocyst abundance tracks cyanobacteria abundance, but in the revised version of manuscript, we also show that heterocyst frequency per filament decreases when biomass increases in November (see figure above).

You need to discuss problems with $N_2$ fixation from covered flasks and still standing flasks with gas injections somewhere in the discussion. "Caveats" with this study.

**Answer**: in the revised manuscript, we have stated that "*Our study, as well as prior work, is based on 24-h incubations, simulating conditions at a fixed depth, which may not be indicative of rates that could be sustained by diazotrophs circulating over a range of depth and light conditions.*" (line 359-361)

**Technical corrections**

Line 33, comma after algae.

**Answer**: added.

Line 39, I think you need to name this 2020a since it the first one appearing?

**Answer**: corrected.

Line 500, Please change to Riemann

**Answer**: done.

---

## Referee Comment (RC4) · Anonymous Referee #1 · 12 Jan 2021

I am pleased to see that you have now met all my comments in a good way. I have no further comments on the manuscript except for that in the last comment you said the N2 fixation rates for the Bothnian Sea in mol instead of what I think should be mmol?

I am also pleased to see that the graphs with biomass/N2fixation/heterocysts now make sense after you found that error in the previous version of your figure.

---

## Author Comment (AC2) · 12 Jan 2021

**Response to Anonymous Referee #1**

**General comments**

**Reviewer comment (RC3):** I am pleased to see that the authors have improved the manuscript according to the comments below. Which includes being open about some caveats, comparison to other studies, some methods that were missing and additional data on heterocyst frequency. However, there is still some minor adjustments I would like to suggest before moving on. See specific comments below.

**Answer:** we acknowledge the reviewer for their positive comments, which are appreciated. Our responses to follow-up comments are provided below.

**Reviewer comment (RC1):** (Something that was surprising to me was how come you didn't find any picocyanobacteria? In Zilius et al. 2020 I interpret it as you had about 20% of the community during summer? Also in Klawonn et al. 2016, colonial picocyanobacterial comprise ca. 5-10% of the cyanobacterial community in terms of carbon. It seems like you sampled on similar locations, maybe even at the same time, as in Zilius et al. 2020 so this needs an explanation. If it has to do with method differences, it needs to be explained or the statement of no picocyanobacterial removed and refer to previous study.

**Answer:** We acknowledge the reviewer for their positive comments. In this study, taxa referred as "colonial picocyanobacteria" by the reviewer were found with microscopy counting, and due to their relatively low contribution (generally <2% of total biomass) they were assigned to "non-$N_2$-fixing cyanobacteria", and thus not further discussed in the submitted manuscript (Fig. 2). In the revised version of our manuscript, we have added information related to cyanobacteria composition and their biomass: "*Non-filamentous colonial cyanobacteria, such as Aphanocapsa spp., Aphanothece spp., Merismopedia spp. and Cyanodictyon spp. exhibited low biomass (< 2% of total) except in June, when their contribution reached 12% at the northern site (Fig. 2). Picocyanobacteria were not detected during the study period at either site." (line 207-210*)

In Zilius et al. 2020, sequences were attributed to picocyanobacteria (not referring here as "colonial picocyanobacteria"). However, a volume of 50 to 70 ml was extracted for further sequencing and only few reads were assigned to picocyanobacteria. This means that picocyanobacteria were rare in this study and that they would not be detected by methods allowing quantification such as flow cytometry or epifluorescence microscopy. Both approaches are complementary and not contradictory since DNA methods can detect rare taxa but do not allow quantification yet.

**Reviewer comment (RC3):** thank you for clarifying this in the revision of the manuscript and for looking further into this by also applying microscopy in addition to flow cytometry. Feel free to also include this extra information so that future readers do not confuse between the groups of colonial vs. free-living picocyanobacteria.

**Answer:** we have added following text: "*Though sequences were also attributed to diazotrophic picocyanobacteria (Synechococcus, Crocosphaera, Rippkaea, and Cyanothece), these were not detected with flow cytometry, suggesting low abundance.*" (line 298-300).

**Reviewer comment (RC1):** I am also a bit concerned about the method you use for measuring $N_2$-fixation with injection of gas rather than pre-dissolved. I think this might cause an underestimation. Also the fact that you run 24 h incubations probably lead to underestimations of $N_2$-fixation per h since they do less in the night when its dark (1.8 times less; Klawonn et al. 2016). I think a potential underestimation should be discussed and rates presented as per day since this is what you measure.

**Answer:** Regarding the issue of hourly vs. daily rates of fixation, we agree with the reviewer's point that rates are likely to vary on a diel cycle (being lower at night). Therefore our diel incubations conducted under natural (outdoor) light conditions are more suitably expressed as daily rates than hourly rates since they are representative of both light and dark cycles. In the revised manuscript, we present daily values in figures and text.

With regards to methodology, we agree that there has been some debate about using the bubble method for $N_2$ fixation measurements (Mohr et al., 2010; Großkopf et al., 2012; White et al., 2020), but recent work (Wannicke et al., 2018) demonstrated that underestimation of rates is negligible (<1%) for incubations lasting 12–24 h. In the submitted version we have argued our choice for incubation duration: "*As the isotopic equilibration takes up to several hours (Mohr et al., 2010), we incubated the samples for 24 h, thus minimizing equilibration effects (Mulholland et al., 2012; Wannicke et al., 2018.*" *(line 136-138).* Eventually, our used technique avoids to have low labelling (percentage label should be between 5-10%) as the labelled seawater method often results in low quantities of $^{15}N_2$ gas in the water (e.g. Klawonn et al. (2015) had only 1% label in their experiment).

**Reviewer comment (RC3):** I am not fully sure if the last sentence about the optimum labelling ranges is within the revision or only a reply to the comment. But if it is in the text, do you have a reference for the mentioned optimum labelling percentages? I do not see any problem with having 1 % labelling when using the dilution approach (which you refer to in Klawonn et al.) since you then have this labelling in the whole flask already from the beginning, and as long as you can trace it and measure the final concentration. In case there is no suitable reference for the optimum ranges, and you want to mention it in the manuscript, I would suggest that you rephrase it to only mention that this range of percentages works well when using the bubble method and not compare to Klawonn et al. as an example of where it has not worked well(?) since I do not know if you have proof of this, and its slightly different methods. Further, I think it is great that you changed to 24 h values since this is what you measured and it includes both day and night.

**Answer:** our reply with regard to $^{15}N$ labelling was provided only in the "Responses to Reviewer". Since we did not measure the percentage of labelling, this information will not be included in "Material and Methods".

**Reviewer comment (RC1):** The theory of underestimation is further supported by that you have 1000-3000 µg cyanobacterial C per L and 120-200 nmol N2 fixation per h as compared to Klawonn 2016 where 100 µg cyanobacteria per L performed 80 nmol $N_2$ fixation per h. Why do you think you have so low rates as compared to your biomass? Can it be P limitation?

**Answer:** We agree with the reviewer's point that P limitation may play a role in limiting N fixation in the Curonian Lagoon. In a prior study, it was shown that P additions stimulated growth rates of $N_2$ fixing cyanobacteria from the Curonian Lagoon (Pilkaitytė and Razinkovas, 2007*).* Likewise, addition of P stimulated diazotrophic community resulting in elevated $N_2$ fixation rates (Moisander et al. 2007). We may expect that dissolved P was limiting, which constrained $N_2$ fixation during summer. Thus, we suppose that DIP release from sediment and higher biomass of *Aphanizomenon* and diazotrophic activity frequently observed in the end of summer (Zilius et al. 2014, 2018) are not coincidence but rather consequence of increased P availability. We have modified the Discussion to address this point "*The proliferation of heterocystous cyanobacteria in the Curonian Lagoon is favoured by P (Pilkaitytė and Razinkovas, 2007), which is released from sediments, particularly when bloom conditions result in high water column respiration and transient (night-time) depletion of oxygen (Petkuviene et al., 2016; Zilius et al., 2014). Moisander et al. (2007) demonstrated that P can enhance diazotrophic activity of heterocystous cyanobacteria in microcosms. Release of dissolved P from sediments may in turn enhance rates $N_2$ fixation resulting in a positive feedback for cyanobacteria bloom development."* (line 299-304)

**Reviewer comment (RC3):** In addition to stating that they are favoured by it you should maybe also say that this might be why they are performing compare-wise lower $N_2$ fixation in comparison to other regions of the Baltic Sea and provide some reference to P concentrations in the lagoon during summer. Maybe they also have enough of other sources of N, such as ammonium, to support some of their needs?

**Answer**: thanks for the comment, which is opening a new question why so high biomass of cyanobacteria can be present in the Curonian Lagoon. We rephrased this section in the revised version of manuscript: "*Measured summer DIP concentration (0.3 µM) in the Curonian Lagoon was similar to that in other Baltic coastal sites (e.g. Klawonn et al., 2016), suggesting that higher biomass might be also supported by higher N availability. A recent study by Broman et al. (submitted) suggests that $N_2$ fixation in the lagoon satisfies only 13% of N demand for phytoplankton. Thus, other internal sources such as N release from sediment and mineralization in the water column are important to meeting algal N demands.*" (line 316-320).

**Specific comments**

**Reviewer comment (RC1):** Line 45-46, when they are dead I guess? Maybe clarify that this is when they are detritus on the bottom.

**Answer**: we assume that respiration of living cells rather detritus cause bottom hypoxia. During summer blooms, when plankton (mainly cyanobacteria and heterotrophic bacteria) respiration exceed diffusive oxygen supply to deeper layer, benthic community eventually depletes oxygen from adjacent bottom. We have clarified sentence and it reads now "*Large blooms of living cyanobacteria are associated with high oxygen demand in the water column, which results in transient (night-time) bottom hypoxia and enhances the release of dissolved inorganic phosphorus (DIP) from sediments (Petkuviene et al., 2016; Zilius et al., 2014).*" (line 45-47)

**Reviewer comment (RC3):** Thank you for the explanation. Although this might be part of the problem, I think a majority of the oxygen on the bottom is consumed as they die and need to be degraded. I think you should add a sentence and a reference to this as well if you mentioned the above statement. See for example Conley et al. 2009 on Hypoxia in the Baltic Sea.

**Answer**: we agree that both respiration by algae and heterotrophic respiration of their decomposing remains contribute to water column oxygen demand. We have modified the text accordingly: "*Large blooms of living cyanobacteria are associated with high oxygen demand in the water column, which we attribute to heterotrophic respiration of algal biomass and respiration by the algae themselves. High oxygen demand results in transient (night-time) hypoxia and enhances the release of dissolved inorganic phosphorus (DIP) from sediments (Petkuviene et al., 2016; Zilius et al., 2014).*" (line 46-48).

**Reviewer comment (RC1):** Line 120, I think it would be good to provide heterocysts per number of vegetative cells as well since they change in density over the season (Svedén et al. 2015).

**Answer**: following the reviewer's suggestion, we added estimates of heterocyst frequency per cyanobacteria filament in the revised version of the manuscript, see updated Figure 6. In the text, we have added following information:

„*The number of heterocysts (cell $L^{-1}$) and their frequency per millimeter of filament ($mm^{-1}$) was also determined.*" (line 121-122)

„*Total heterocyst frequency per filament was higher at the beginning of summer (up to 15 $mm^{-1}$) at both sites, and gradually declined afterwards (Fig. 6c, d).*" (line 249-250)

**Reviewer comment (RC3):** Thank you for including heterocysts per filaments, I think this is more informative than just heterocysts per ml. However, I think you should be more specific. In the southern site the numbers were highest in June and August/September, so actually two peaks, and in the northern sites they peaked in June/July and August depending on the species, so not only early in the summer and declining as the sentence reads now.

**Answer:** thanks for comment. In the revised version we have specified temporal patterns in heterocyst abundance: "*Heterocyst frequency per filament showed distinctive temporal patterns between the studied sites depending on the species (Fig. 6c, d). At the southern site, two peaks up to 8.0 mm$^{-1}$ in heterocyst frequency of both species was observed during June–September. Whereas heterocyst frequency at the northern site remained quite high through summer primarily contributed by A. flosaque (~10 mm$^{-1}$), later followed by Dolichospermum spp. (~8 mm$^{-1}$).*" (line 254-258).

**Reviewer comment (RC1):** Line 181, why linear regression and not correlations? Don't you expect both of them to be interdependent rather than one dependent?

**Answer:** while we consider that correlation coefficient represents the direction and strength of the relationship between chlorophyll and $N_2$ fixation rates, the regression coefficient determines the effect of chlorophyll a (independent variable) on the $N_2$ fixation (dependent variable), and determine the explained variation.

**Reviewer comment (RC3):** I guess this is fine as long as you state this as your tested hypothesis in the methods.

**Answer:** we appreciate that the two variables may be considered inter-dependent (i.e., blooms of diazotrophic cyanobacteria may result in higher rates of $N_2$ fixation, and increased rates of $N_2$ fixation may lead to expansion of cyanobacteria blooms). Our choice of x and y variables is dictated by the fact that Chl-a may be derived via remote sensing and thereby permit estimation of $N_2$ fixation over large spatial and temporal scales. From a modelling perspective, the converse is unlikely to be as useful (i.e., estimating Chl-a from $N_2$ fixation). The regressions models relating $N_2$ fixation to Chl-a simply serve to parameterize the needed variables (slope and intercept) and are not meant to test a hypothesis. Text modified as: "*Estimates of $N_2$ fixation were derived for each of the grid cells based on satellite-derived Chl-a and regression models relating $N_2$ fixation measurements to concurrent in situ measurements of Chl-a (regressions provided in Results). The linear regressions served to parameterize the model components (slope and intercept).*" (line 188-189).

**Reviewer comment (RC1):** Figure 6, do you mean "per ml" with per mil? How come there is so many heterocysts in November in the southern station but almost no $N_2$ fixation nor cyanobacteria biomass at that time? In contrast the highest number of both $N_2$ fixation and heterocysts numbers correlates for the northern station. This needs to be discussed. Also, It would be good to also have heterocysts per filaments/vegetative cells here to see how it changes over the season (*Aphanizomenon* heterocyst density varies with season; Svedén et al. 2015).

**Answer**: "per mil" means delta units that are expressed in molecules per thousand, but for convenience we have change to "‰".

In revised version, we also provided heterocyst frequency, which better corresponded to $N_2$ fixation dynamic. The updated Figure 6 shows that patterns in heterocyst frequency was relatively low in November coinciding to decreased $N_2$ fixation rates. We assume that the October-November period represents the decline of the cyanobacteria bloom. In the submitted version we discussed that "*Results from this, and a prior study (Zilius et al. 2018), show that despite a high abundance of A. flosaquae at the end of fall, heterocyst frequency, and thus $N_2$ fixation rates declined substantially*

*when water temperature dropped below 15 °C. Zakrisson et al. (2014) suggested that temperature controls the enzymatic activity of nitrogenase, which directly regulates the intensity of $N_2$ fixation in filaments.*" (line 363-367)

**Reviewer comment (RC3):** I am still a bit perplexed about how chl a of cyanobacteria I November at the southern site is very low, the heterocysts per filaments is very low but the heterocysts per L is very high? I understand that $N_2$ fixation goes down with temperature, but how can it be so many heterocysts when there is no biomass?

**Answer:** we checked twice our dataset and manuscript, and found that the last bar, representing November in Figure 2a, has changed colour when converting from text file to PDF, which sometimes happens. The correct figure shows that Chl-a in surface layer was high as well as biomass of $N_2$-fixing cyanobacteria and absolute heterocystous number per litre.

[Figure]

**Figure 2:** Phytoplankton and bacteria biomass at southern (a, b) and northern (c) sites in the Curonian Lagoon during 2018. Chlorophyll-a concentrations are mean values and standard error (some error bars not visible) based on three replicates.

**Reviewer comment (RC1):** Lines 260-261, how can you use this relationship when it was not significant, then there is no relationship?

**Answer**: The regression is marginally significant (p = 0.08) and has a reasonable $R^2$ value (0.52). Therefore, we felt that this model provided the best means for estimating bottom layer $N_2$ fixation. We also note that the bottom layer accounts for a relative small proportion of the lagoon's volume and an even smaller proportion of $N_2$ fixation (since surface rates are 7x higher). Therefore, our

whole-lagoon estimates of $N_2$ fixation are not highly sensitive to assumptions about bottom water rates.

**Reviewer comment (RC3):** Is this caveat explained also in the manuscript?

**Answer:** this caveat was added in the text "*We benefitted from prior work deriving Chl-a estimates from satellite images and their calibration to in situ measurements (Bresciani et al., 2014), but the success of the approach largely relied on the fact that heterocystous cyanobacteria dominated the summer–fall phytoplankton community of the lagoon, which provided a significant relationship between $N_2$ fixation and in situ Chl-a in surface layer. The regression model for estimating bottom layer $N_2$ fixation was marginally significant, and therefore we felt that the application of this model to deriving whole-water column rates was warranted. Whole-lagoon estimates were not highly sensitive to assumed rates in the bottom layer because this layer accounts for a relative small proportion of the lagoon's volume and because measured $N_2$ fixation rates in the bottom layer were 7 times lower than the surface.*" (line 331-338).

**Reviewer comment (RC1):** Lines 309 and below, can you also put these areal $N_2$-fixation estimates into perspective to other studies for the region? For example, Klawonn et al. 2016 and Olofsson et al. 2020 as well as references there in.

**Answer:** thanks for suggestion. We have put our estimates in the context of the Baltic region: "*These estimates reveal that summer $N_2$ fixation rates are slightly lower in the coastal site of SW Baltic (3.6 ± 2.6 µmol $m^{-2}$ $d^{-1}$; Klawonn et al., 2016), but higher than those found in the Great Belt (~ 1 mmol $m^{-2}$ $d^{-1}$; Bentzon-Tilia et al., 2015), Baltic Proper (0.4 ± 0.1 mmol $m^{-2}$ $d^{-1}$; Klawonn et al., 2016), and Bothnian Sea (0.6 ± 0.2 mol $m^{-2}$ $d^{-1}$; Olofsson et al. 2020b).*" (line 328-331)

**Reviewer comment (RC3):** Please use the same units across all studies so its easier for the reader to compare. Maybe you need to formulate the sentence a bit clearer: "These estimates reveal that summer $N_2$ fixation rates are slightly lower in the Curonian Lagoon as compared to those measured at a coastal site of…".

**Answer**: we apologise for different units as it is typesetting mistake. The corrected version reads: "*These estimates reveal that summer $N_2$ fixation rates are slightly lower in the Curonian Lagoon as compared to those measured at a coastal site of SW Baltic (3.6 ± 2.6 mmol $m^{-2}$ $d^{-1}$; Klawonn et al., 2016), but higher than those found in the Great Belt (~ 1 mmol $m^{-2}$ $d^{-1}$; Bentzon-Tilia et al., 2015), Baltic Proper (0.4 ± 0.1 mmol $m^{-2}$ $d^{-1}$; Klawonn et al., 2016), and Bothnian Sea (0.6 ± 0.2 mol $m^{-2}$ $d^{-1}$; Olofsson et al. 2020b).*" (line 350-353).

---

## Author Comment (AC3) · 12 Jan 2021

We acknowledge the reviewer for their positive comments. The reviewer is right for N2 rates in the Bothnian Sea, which should be in mmol N m-2 d-1. We have corrected this typesetting mistake.
* * *

---

## Referee Comment (RC5) · Anonymous Referee #2 · 2 Feb 2021

1. General comments (an initial paragraph or section evaluating the overall quality of the preprint) Overall, this is a very nicely written paper that integrates remote sensing data with empirical biogeochemical and biological data to estimate ecosystem-scale N-fixation in an oligohaline coastal lagoon. Using remote sensing data to study processes such as N-fixation, given the good empirical relationship between N-fixation and chl-a in the late summer, is a very nice application of these data in coastal systems. As the authors point out, blooms of N-fixing cyanobacteria can significantly alter the N-budgets of enclosed coastal water bodies. Importantly, this can lead to these systems serving as 'sources' of N to the coastal ocean rather than serving as reactors for removing DIN via denitrification.

[Figure]

One concern I have is the spatial distribution of the water sampling locations. I agree that using the remote sensing approach appears to provide much more resolved estimates of N-fixation (this ms) than simply scaling up from the two sample locations (Line 310-312). Still, the entire southern half of the lagoon was not sampled. For example, it is noted at Line 78 in this ms that, "Longer water residence time in the southern lagoon provides favorable conditions for cyanobacteria bloom development (Bartoli et al., 2018)." Without actually measuring N-fixation rates vs chl-a concentrations at those southern areas (which could differ if the phyto community composition differs), there is still uncertainty about whether or not the remote-sensing based approach is yielding biased results in those southern reaches. This is particularly true because most of the high N-fixation rate hotspots in Figure 8 are further south than the 'southern' sampling location. It seems unlikely that this particular concern can be addressed using the same dataset but it is an important caveat that should be acknowledged. If the authors have evidence that the phytoplankton community in the southern part of the lagoon is the same as the community in the middle of the lagoon (i.e., the 'southern' sampling location) either from previous literature or their own unpublished work, then this would be an important pattern to note for readers.

The methods are very sparse for the TN riverine data collection. While not a central part of the analysis, these data are used to place the remote sensing results in an ecosystem context and are therefore important to the manuscript. In the text, reference is made to a previous paper rather than providing methods, but in the referenced paper (Zilius et al. 2018), the methods reported in that paper are limited to the following: "For the mass balance analysis, water samples were collected at the inflow (Nemunas River) and outflow (Klaipeda Strait) of the lagoon, and from an off-shore site in the Baltic Sea (55°55′13.1"N and 21°02′39.4"E), to estimate riverine inputs, lagoon export, and marine inputs, respectively (Fig. 1). Samples were collected monthly at each of the sites from December 2014 to November 2015, except at the inflow site (Nemunas) where additional samples were obtained (at 1–2 week intervals) during the period of highest discharge (January–April)." (Zillius et al. 2018) It is important to see

some additional details, even if they are only provided in the Supplemental file. Were samples collected monthly or at higher resolution at certain times of the year (as in Zilius et al. 2018)? Was there any effort to collect during average flow conditions? Where were the samples collected – mid-stream in the river, or from the shore? Is the collection location the same location referenced in the Zilius et al. 2018 paper?

2. Specific comments Figure 1. Please provide definitions for abbreviations (RUS, LT) and increase font size on some of the smaller figure elements such as the scale bar. There is a gray rectangle just above the Nemunas River. . .is that meant to be there and if so, what is it? Please show the river sampling location on the map.

Line 114 – please provide a long-term estimate of d15N analytical precision for the UC Davis facility. They should have these numbers readily available. Otherwise, you could also report summary statistics on sample duplicates that were (presumably) interspersed with the submitted samples.

Figure 3 – it is confusing to list Anabaena in the figure while referring to it as Dolichospermum in the text. There is a note in the figure legend that the two are the same but why not simply use Dolichspermum in the figure (or at least an abbreviation)?

Figure 4 – are the southern site values averaged between surface and bottom or are these only surface (or bottom) values? Can you please clarify in the figure legend? Line 346-360 – also see papers by Karlson et al. (2015), Woodland, Cook and others (2013, 2014) for evidence of diazotrophic N from cyanobacteria contributing to brackish food webs

3. Technical corrections Line 182 – what do you mean by 'process' here? Is that word out of place or does it reference to a specific type of measurement taken in the surface and bottom waters? Can you please rephrase to make this more interpretable?

Line 248 – add a comma after '0.88'

Line 249 – add a space between '=' and '0.07'

Line 356-357 – replace 'their' with 'these blooms to have a' or something similar. The current phrasing is awkward

References: Karlson, A.M.L., Duberg, J., Motwani, N.H. et al. Nitrogen fixation by cyanobacteria stimulates production in Baltic food webs. AMBIO 44, 413–426 (2015). https://doi.org/10.1007/s13280-015-0660-x

Woodland RJ, Holland DP, Beardall J, Smith J, Scicluna T, Cook PLM (2013) Assimilation of Diazotrophic Nitrogen into Pelagic Food Webs. PLoS ONE 8(6): e67588. https://doi.org/10.1371/journal.pone.0067588

Woodland, R.J. and Cook, P.L.M. (2014), Using stable isotope ratios to estimate atmospheric nitrogen fixed by cyanobacteria at the ecosystem scale. Ecological Applications, 24: 539-547. https://doi.org/10.1890/13-0947.1

---

## Author Comment (AC4) · 3 Feb 2021

**Response to Anonymous Referee #2**

**General comments**

Overall, this is a very nicely written paper that integrates remote sensing data with empirical biogeochemical and biological data to estimate ecosystem-scale N fixation in an oligohaline coastal lagoon. Using remote sensing data to study processes such as N-fixation, given the good empirical relationship between N-fixation and chl-a in the late summer, is a very nice application of these data in coastal systems. As the authors point out, blooms of N-fixing cyanobacteria can significantly alter the N-budgets of enclosed coastal water bodies. Importantly, this can lead to these systems serving as 'sources' of N to the coastal ocean rather than serving as reactors for removing DIN via denitrification.

**Answer:** we acknowledge the reviewer for their positive comments, which are appreciated.

One concern I have is the spatial distribution of the water sampling locations. I agree that using the remote sensing approach appears to provide much more resolved estimates of N-fixation (this ms) than simply scaling up from the two sample locations (Line 310-312). Still, the entire southern half of the lagoon was not sampled. For example, it is noted at Line 78 in this ms that, "Longer water residence time in the southern lagoon provides favorable conditions for cyanobacteria bloom development (Bartoli et al., 2018)." Without actually measuring N-fixation rates vs chl-a concentrations at those southern areas (which could differ if the phyto community composition differs), there is still uncertainty about whether or not the remote-sensing based approach is yielding biased results in those southern reaches. This is particularly true because most of the high N-fixation rate hotspots in Figure 8 are further south than the 'southern' sampling location. It seems unlikely that this particular concern can be addressed using the same dataset but it is an important caveat that should be acknowledged. If the authors have evidence that the phytoplankton community in the southern part of the lagoon is the same as the community in the middle of the lagoon (i.e., the 'southern' sampling location) either from previous literature or their own unpublished work, then this would be an important pattern to note for readers.

**Answer:** we agree that whole lagoon sampling would be an ideal, but access to the southern region, which is located within Russian territorial waters, is problematic. Here, we improve on our ability to scale up these measurements by using remote sensing of Chl-a to infer spatial and temporal variation in $N_2$ fixation. Our whole-lagoon estimates are based on data collected at stations within the northern and central portions of the lagoon, as access to the southern region is problematic. Hydrodynamic modeling studies have shown that water renewal times in the central and southern portions of the lagoon are comparable (Umgiesser et al. 2016). Monitoring data suggest that Chl-a and phytoplankton community composition is similar in the central and southern regions (Bresciani et al. 2014; Semenova and Dmitrieva 2011). Therefore, we felt it was appropriate to derive whole-lagoon estimates of N fixation based on in situ measurements from these two sites.

References:

Bresciani, M., Adamo, M., De Carolis, G., Matta, E., Pasquariello, G., Vaičiūtė, D., and Giardino, C.: Monitoring blooms and surface accumulation of cyanobacteria in the Curonian

Lagoon by combining MERIS and ASAR data. Remote Sens. Environ., 146, 124–135, doi:10.1016/j.rse.2013.07.040, 2014.

Umgiesser, G., Zemlys, P., Erturk, A., Razinkova-Baziukas, A., Mežinė, J., and Ferrarin, Ch.: Seasonal renewal time variability in the Curonian Lagoon caused by atmospheric and hydrographical forcing. Ocean. Sci., 12, 391–402, doi:10.5194/os-12-391-2016, 2016.

Semenova A. S., and Dimitrieva O. A.: Spatial and temporal aspects of toxic effect of harmful algae on zooplankton in the Curonian Lagoon (the Baltic Sea) in New series 1(4) by Trudy AtlantNIRO. AtlantNIRO, Kaliningrad, RUS, 56–69, 2017.

The methods are very sparse for the TN riverine data collection. While not a central part of the analysis, these data are used to place the remote sensing results in an ecosystem context and are therefore important to the manuscript. In the text, reference is made to a previous paper rather than providing methods, but in the referenced paper (Zilius et al. 2018), the methods reported in that paper are limited to the following: "For the mass balance analysis, water samples were collected at the inflow (Nemunas River) and outflow (Klaipeda Strait) of the lagoon, and from an off-shore site in the Baltic Sea (55◦55âAš13.1"N and 21 ˇ ◦02âAš39.4"E), to estimate riverine inputs, lagoon export, and December 2014 to November 2015, except at the inflow site (Nemunas) where additional samples were obtained (at 1–2 week intervals) during the period of highest discharge (January–April)." (Zillius et al. 2018) It is important to see some additional details, even if they are only provided in the Supplemental file. Were samples collected monthly or at higher resolution at certain times of the year (as in Zilius et al. 2018)? Was there any effort to collect during average flow conditions? Where were the samples collected – mid stream in the river, or from the shore? Is the collection location the same location marine inputs, respectively (Fig. 1). Samples were collected monthly at each of the sites from referenced in the Zilius et al. 2018 paper?

**Answer:** we have provided some additional details of methodology to reduce reliance on the Zilius et al. 2018 paper: "*We also monitored total nitrogen (TN) concentrations in the Nemunas River (Fig. 1) to derive riverine N loads for comparison with atmospheric N inputs via $N_2$ fixation*. River samples were collected twice monthly during peak discharge (January-April) and monthly throughout the rest of the year (16 collections). *Water samples (2 L) were collected in triplicate, integrating the whole water column with repeated Ruttner bottle sampling at the surface (0.4 m depth) and bottom layers (3.0 m depth) as described in Vybernaite-Lubiene et al. (2018). Integrated water samples were transferred to opaque bottles, cooled with ice packs, and transported to the laboratory within the hour for subsequent analyses (see section 2.3 for details). Riverine N concentrations were used in combination with daily discharge measurements (provided by Lithuanian Hydrometeorological Service) to derive monthly N loads to the lagoon as previously described in Zilius et al. (2018).*" (line 98-105).

**Specific comments**

Figure 1. Please provide definitions for abbreviations (RUS, LT) and increase font size on some of the smaller figure elements such as the scale bar. There is a grey rectangle just above the Nemunas River. Is that meant to be there and if so, what is it? Please show the river sampling location on the map.

**Answer**: thanks for suggestions. We have updated figure accordingly.

Line 114 – please provide a long-term estimate of d15N analytical precision for the UC Davis facility. They should have these numbers readily available. Otherwise, you could also report summary statistics on sample duplicates that were (presumably) interspersed with the submitted samples.

**Answer**: we have added missing information "*The long-term standard deviation is <0.3 ‰ for $\delta^{15}N$.*" (line 128-129)

Figure 3 – it is confusing to list *Anabaena* in the figure while referring to it as *Dolichospermum* in the text. There is a note in the figure legend that the two are the same but why not simply use *Dolichspermum* in the figure (or at least an abbreviation)?

**Answer**: thanks for suggestion. We have corrected figure accordingly (see below).

[Figure]

**Figure 3:** Principal coordinate biplots generated on Euclidean distances of normalized and forth-root transformed nutrient concentrations (DOC, $NH_4^+$, $NO_2^-$, $NO_3^-$, DON, DIP, DOP, and DIN:DIP). Overlaid vectors show individual chemical variables (those significantly correlating with either of the two primary axes, with Pearson correlations > 0.5) and plankton community biomass (*Aphanizomenon, Dolichospermum,* non-$N_2$-fixing cyanobacteria and heterotrophic bacteria).

Figure 4 – are the southern site values averaged between surface and bottom or are these only surface (or bottom) values? Can you please clarify in the figure legend?

**Answer**: revised caption is "*Figure 4: Temporal patterns in temperature, dissolved organic carbon (a, b), dissolved and organic nitrogen (c, e), phosphorus (e, f), and DIN:DIP ratios (g, h) at southern (surface layer; left panel) and northern (right panel) sites in the Curonian*

*Lagoon during 2018 (error bars denote standard error based on 3 replicates; some not visible).*"

Line 346-360 – also see papers by Karlson et al. (2015), Woodland, Cook and others (2013, 2014) for evidence of diazotrophic N from cyanobacteria contributing to brackish food webs.

**Answer**: thanks for suggestion. We have included these references in manuscript.

**Technical corrections**

Line 182 – what do you mean by 'process' here? Is that word out of place or does it reference to a specific type of measurement taken in the surface and bottom waters? Can you please rephrase to make this more interpretable?

**Answer**: this is a redundant and has removed from the text.

Line 248 – add a comma after '0.88'

**Answer**: Done.

Line 249 – add a space between '=' and '0.07

**Answer**: Done.

Line 356-357 – replace 'their' with 'these blooms to have a' or something similar. The current phrasing is awkward.

**Answer**: this comment is not clear as indicated line reads "…abundance of *Microcystis* spp. and *Planktotrix agardhii*. Measured low $\delta^{15}N$ values (0.5 ± 0.2 ‰) in suspended living material suggest that fixed N can temporally support most of the nutritional needs for plankton (bacteria + phytoplankton) growth".

We may think that the reviewer had indicated line 365-367, "*Since intensifying blooms of cyanobacteria have already been observed in coastal areas of the Baltic Sea (Olofsson et al., 2020b), we may expect their stronger effect on ecosystem functioning in future*". We have rephrased this sentence, "*Since intensifying blooms of cyanobacteria have already been observed in coastal areas of the Baltic Sea (Olofsson et al., 2020a), we may expect these blooms to have a stronger effect on ecosystem functioning in future*".

---

## Author Response (AR2)

Dear Editor,

I acknowledge the Editor for positive comments, which are appreciated.

**Question**. Given the shallow nature of the lagoon, do you think higher respiration might be another mechanism driving P release rather than just anoxia?

**Answer:** I am not sure I am understanding the question that is being raised here. If I am right the Editor asking whether greater P flux from sediments during transient night-time oxygen depletion associated with cyano blooms is due to (a) more favorable conditions for P release (due to anoxia), or (b) higher rates of P remineralization (via sediment OM respiration)?

Given the fact that lagoon is mostly freshwater, the sulphate reduction within sediment should be limited during hypoxic/anoxic events. Therefore, I think that ferric iron reduction via respiratory pathways is likely responsible mechanism for DIP liberation and further release from sediment to bottom water (Zilius et al. 2015).

However, I cannot neglect that DIP production within sediments also depends on aerobic respiration as shows correlation between $O_2$ and DIP fluxes ($r_s=0.37$, $p<0.001$, $n=135$) from our long-term benthic flux measurements in muddy area. While our recent study reveals that due to positive buoyancy cyanobacteria accumulation in sediments is likely limited (Zilius et al. 2018)

References

Zilius, M., et al. 2015. Phosphorus mobility under short-term anoxic conditions in two shallow eutrophic coastal systems (Curonian and Sacca di Goro lagoons). Estuarine, Coastal and Shelf Science 164: 134-146.

Zilius, M. et al. 2018. The influence of cyanobacteria blooms on the attenuation of nitrogen throughputs in a Baltic coastal lagoon. Biogeochemistry, 141(2), 143–165, doi:10.1007/s10533-018-0508-0.

**Comment**: Line 391-392 please check ref dates:

**Answer**: done. Here we cited:

Karlson, A. M. L., Duberg, J., Motwani, N. H., et al.: Nitrogen fixation by cyanobacteria stimulates production in Baltic food webs. Ambio, 44, 413–426, doi:10.1007/s13280-015-0660-x, 2015.

Woodland, R. J., Holland, D.P., Beardall, J., Smith, J., Scicluna, T., and Cook, P. L. M.: Assimilation of diazotrophic nitrogen into pelagic food webs. PLoS ONE, 8(6), e67588, doi:10.1371/journal.pone.0067588, 2013.

I would like kindly to know if my reply met raised question.

Sincerely,

Mindaugas Zilius